# Insurance loss model vs meteorological loss index – How comparable are their loss estimates for European windstorms?

Julia Moemken[1*], Inovasita Alifdini[1*], Alexandre M. Ramos[1], Alexandros Georgiadis[2], Aidan Brocklehurst[2], Lukas Braun[3], Joaquim G. Pinto[1]

[1]Institute of Meteorology and Climate Research Troposphere Research (IMKTRO), Karlsruhe Institute of Technology (KIT), Karlsruhe, Germany
[2]Aon Impact Forecasting, London, UK
[3]Aon Impact Forecasting, Prague, Czech Republic

[*]These authors contributed equally to this work.

Correspondence to: Julia Moemken (julia.moemken@kit.edu)

**Abstract.** Windstorms affecting Europe are among the natural hazards with the largest socio-economic impacts. Therefore, many sectors like society, economy or the insurance industry are highly interested in reliable information on associated impacts and losses. In this study, we compare – for the first time – estimated windstorm losses using a simplified meteorological loss index (LI) with losses obtained from a complex insurance loss (catastrophe) model, namely the European Windstorm Model of Aon Impact Forecasting. To test the sensitivity of LI to different meteorological input data, we furthermore contrast LI based on the reanalysis dataset ERA5 and its predecessor ERA-Interim. We focus on similarities and differences between the datasets in terms of loss values and storm rank for specific historical storm events in the common reanalysis period across 11 European countries.

Our results reveal higher LI values for ERA5 than for ERA-Interim for all of Europe (by roughly a factor of 10), coming mostly from the higher spatial resolution in ERA5. The storm ranking is comparable for Western and Central European countries for both reanalyses, confirmed by high correlation values between 0.6 and 0.89. Compared to Aon's Impact Forecasting model, LI ERA5 shows comparable storm ranks, with correlation values ranging between 0.45 and 0.8. In terms of normalized loss, LI exhibits overall lower values and smaller regional differences. Compared to the market perspective represented by the insurance loss model, LI seems to have particular difficulty in distinguishing between high impact events at the tail of the wind gust distribution and moderate impact events. Thus, the loss distribution in LI is likely not steep enough and the tail is probably underestimated. Nevertheless, it is an effective index that is suitable for estimating the impacts and ranking storm events, precisely because of its simplicity.

## 1 Introduction

In Central and Western Europe, windstorms are among the major natural hazards. They regularly lead to high economic and insured losses (Munich Re, 2022), causing damage to natural and human-made environments like infrastructure, buildings,

forestry and agriculture (Mitchell-Wallace et al., 2017; Pinto et al., 2019; Gliksman et al., 2023). In 2022, losses from European windstorms were well above average, with insured losses of \$5.7 billion and economic losses of \$7.5 billion, respectively (Aon, 2023a). In fact, European windstorms were among the five largest weather-related perils in 2022 (Swiss Re, 2023). The high losses were mainly caused by the windstorm series Ylenia-Zeynep-Antonia[1] (international: Dudley-Eunice-Franklin) in February 2022, which resulted in insured losses of \$4.7 billion (Aon, 2023a). The storm series affected the British Isles and continental Europe (Mühr et al., 2022), with highest losses in Germany, the Benelux countries, the UK and France (PERILS, 2023). While from the market perspective there is a large number of impactful storms in recent decades, there is no clear trend and mostly decadal variability apparent over the last hundred years in meteorological terms (see review in Feser et al. (2015)).

For the insurance industry, but also for society and economy, it is crucial to assess the wind-related risk, determine the return periods of historical storms and to forecast the impacts of extreme storms in order to adapt to and mitigate windstorm losses (Mitchell-Wallace et al., 2017; Pinto et al., 2019; Merz et al., 2020; Raschke, 2022; Gliksman et al., 2023). In this context, risk is usually defined as the interaction of hazard, exposure and vulnerability (e.g. IPCC, 2022). The hazard component is defined as the occurrence of a natural event (in our case a windstorm), the exposure component represents the presence of people/livelihoods/ecosystems or economic/social assets, and the vulnerability component describes the disposition to be affected (IPCC, 2022). The information on windstorm risk and associated losses is provided by various types of datasets (Gliksman et al., 2023; Moemken et al., 2024), both for present and future climate conditions. These datasets do not always account for all three risk components. Meteorological indices/storm severity indices (e.g. Klawa and Ulbrich, 2003) combine meteorological variables and insurance aspects, usually only considering the hazard component. The more complex storm loss models - developed, among others, by the insurance industry (catastrophe modelling) – consider all three risk components. These models relate meteorological wind data to actual building damage data, using so-called damage functions that define the relationship between wind and damage (Prahl et al., 2015; Gliksman et al., 2023). Commonly used damage functions assume either a power law or an exponential form.

Moemken et al. (2024) recently compared several examples of loss datasets for windstorms across Europe, including a natural hazard database, insurance loss reports and various meteorological indices. Focusing on storm numbers and the ranking of specific storm events, they conclude that the datasets provide different perspectives on windstorm impacts and suggest that a combination of different types of datasets might be used to assign an uncertainty range to windstorm losses. A recent review paper by Gliksman et al. (2023) discusses open research questions related to damage from European windstorms. One raised issue is the lack of a clear methodology to select the most suitable index to assess windstorm losses for both present and future climate conditions. Moreover, loss calculations are affected by uncertainties, for example related to the used (meteorological) input data. They further point out that there is a need for a thorough comparison between meteorological loss indices and catastrophe models (used in insurance) to better understand loss estimates from different perspectives.

In this study, we try to answer two of these questions, namely:

---

[1] Storm names as given by the Freie Universität Berlin (https://www.wetterpate.de/namenslisten/tiefdruckgebiete/index.html; in German) and used by the German Weather Service (DWD).

- How sensitive are the loss estimates of a meteorological index to the meteorological input data?

- How comparable are windstorm loss estimates from this meteorological index and an insurance loss model?

With this aim, we first calculate the Loss Index (LI) by Pinto et al. (2012) in the adaptation of Karremann et al. (2014a) using ERA5 (Hersbach et al., 2020) and its predecessor ERA-Interim (Dee et al., 2011). In a second step, we compare the loss estimates from LI to the output of an insurance loss (catastrophe) model for a set of historical European windstorms. Here, we use, for the first time in a scientific study, the European Windstorm Model of Aon Impact Forecasting (in the following Aon's

IF Euro WS model). We analyse the differences and similarities, focusing on loss values and storm ranks of individual events. The study is restricted to 11 European countries covered by Aon's IF Euro WS model (see Sect. 3.2) and the extended winter season October – March (ONDJFM). For proprietary reasons, we only show country-aggregated and normalized losses. Throughout this study, we use the terms "extreme" and "severe" interchangeably when referring to storm events with high losses.

The paper is organized as follows: Section 2 describes the datasets and Section 3 the methods/models. Section 4 focuses on the sensitivity of LI to different reanalysis datasets, while Section 5 presents the comparison between LI and Aon's IF Euro WS model. Section 6 concludes this paper with a summary and discussion of results.

## 2 Data

### 2.1 Meteorological input data

For the calculation of LI (Sect. 3.1.1), gridded datasets are needed. As we are interested in historical windstorms, we use reanalysis data – namely ERA5 (Hersbach et al., 2020) and its predecessor ERA-Interim (Dee et al., 2011). ERA5 is the latest reanalysis product of the European Centre for Medium-Range Weather Forecast (ECMWF). Wind gust data is available at hourly temporal and 30 km (0.25°) horizontal resolution for the period 1959-2021. For ERA-Interim, wind gust data is available with 3-hourly temporal and 83 km (0.75°) horizontal resolution for the period 1979-2019. In both datasets, wind

gusts are defined as the maximum 3-second wind at 10 m height following the definition of the World Meteorological Organisation (WMO). For both datasets, ECMWF publishes post-processed wind gust, which is the maximum gust computed in every time step following the standard parameterization approach by Panofsky et al. (1977) and Bechthold and Bidlot (2008). We use the datasets in their native resolutions in order to test the sensitivity of LI to the resolution of the input data, considering only the common period 1979-2019. Additionally, we repeated some of the analyses with ERA5 data re-gridded to the coarser

ERA-Interim grid using a conservative remapping.

### 2.2 Insurance data – PERILS

For the insurance perspective of the impacts of the windstorms, we use the PERILS data (https://www.perils.org). PERILS is a joint stock company owned by ten shareholders from the insurance industry, which collects, homogenises and provides aggregated anonymized insurance data for different weather-related perils (see Moemken et al. (2024) for a detailed

description). The data provides for selected events with a sizable financial footprint a market estimation for the loss per country and CRESTA zone (a geographical data aggregation standard used by global insurance industry; www.cresta.org), property premium data per country, and the exposure (total sum of insured property) per country and CRESTA zone. For extratropical windstorms in Europe, PERILS provides data for 12 countries, 11 of which are also covered by Aon's IF Euro WS model, namely Austria, *Belgium*, *Denmark*, *France*, *Germany*, *Ireland*, *Luxembourg*, *the Netherlands*, Norway, *the United Kingdom*, and Sweden. Following Pinto et al. (2012), we additionally focus on the region of Core Europe, which is of special interest for the insurance industry in terms of windstorm risk and consists of those eight countries highlighted in italics. The insurance data is supplied by PERILS on an annual subscription basis. In our study, we use the exposure data of PERILS for the exposure component in the Aon IF model (Sect. 3.2) and the loss data as reference data for some of the analyses.

### 2.3 Storm names

We assign a name to each storm event based on the date of its occurrence, referring to those given by the Freie Universität Berlin and used by the German Weather Service (DWD): https://www.wetterpate.de/namenslisten/tiefdruckgebiete/index.html (in German). For events prior to 1999, we also refer to the Extreme Windstorms Catalogue (XWS) described in Roberts et al. (2014) and the windstorm documentation by Deutsche Rück for the years 1997-2004 (Deutsche Rück, 2005).

## 3 Methods

### 3.1 Meteorological loss index

Meteorological loss indices, also referred to as storm severity indices, are typically used to identify severe windstorms, study their magnitude and likelihood of occurrence, and estimate the associated losses. There exists a wide variety of indices, ranging from more general ones to those targeting specific sectors like forestry, agriculture or transport (see Gliksman et al. (2023) for a detailed overview). The key variable for many of these indices is the daily maximum wind speed or peak wind gust, which is considered as relevant for storm losses (Lamb, 1991; Klawa and Ulbrich, 2003; Leckebusch et al., 2008; Pardowitz et al., 2016). The assumption behind this is that the loss can be primarily attributed to the maximum gust, which causes damage by generating "pressure" on the infrastructure (Klawa and Ulbrich, 2003).

### 3.1.1 Loss Index LI

In our study, we use the Loss Index (LI) by Pinto et al. (2012) in the extended version by Karremann et al. (2014a). LI is built from the widely used storm loss model by Klawa and Ulbrich (2003) and is based on the following assumptions:

- Losses increase with the cube of wind speed/gust (Palutikof and Skellern, 1991; Lamb, 1991), which – from a physical perspective – is proportional to the wind power or the wind kinetic energy flux.

- Infrastructure and other assets are adapted to the local wind conditions. Therefore, it can be assumed that only the top 2% of wind gusts (corresponding to beaufort 8, circa 17-20 m/s) cause damage to buildings (Palutikof and Skellern, 1991). This is taken into account by scaling the daily peak gust with the local 98$^{th}$ percentile.

- In the case that no insurance data is available, population density can be used as a proxy for the insured value (exposure component).

- The (re-)insurance clause for natural hazards is typically 72 hours. This also corresponds to the period during which an average storm crosses Europe and produces damaging winds (Hewson and Neu, 2015).

Hence, LI is calculated as:

$$LI = \sum_{i=1}^{N} \sum_{j=1}^{M} \left( \frac{v_{ij}}{v_{98_{ij}}} \right)^3 * I\left(v_{ij}, v_{98_{ij}}\right) * P_{ij} * L_{ij} \tag{1}$$

with $I\left(v_{ij}, v_{98_{ij}}\right) = \begin{cases} 0 \; for \; v_{ij} < v_{98_{ij}} \\ 1 \; for \; v_{ij} > v_{98_{ij}} \end{cases}$,

$L_{ij} = \begin{cases} 0 \; over \; sea \\ 1 \; over \; land \end{cases}$,

maximum wind gust v in 72 hours at grid point ij, local 98$^{th}$ percentile $v_{98}$, and population density P. Here, we use gridded population density data for the year 2020 at a spatial resolution of 0.25° (see Figure 1a), downloaded from the Centre for International Earth Science Information Network (CIESIN) at Columbia University, USA.

To separate individual events per extended winter season (October – March, ONDJFM), overlapping 72-hour sliding time windows (shifted every 6 hours) are used and the temporal local maximum of each 72-hour time window is analysed (Karremann et al., 2014a). We are particularly interested in extreme storm events. Therefore, we only consider events with LI values above a certain threshold, which corresponds to the selection of an average of five events per season (Pinto et al., 2012; Karremann et al., 2014a). This results in 205 storm events (41 years x 5 events) per dataset (LI ERA5 and LI ERA-Interim).

### 3.1.2 Windstorm footprints

For the hazard component, windstorm footprints are required. Following the WMO and Haylock (2011), the footprint is defined as the percentage of wind gust values that exceed the local 98$^{th}$ percentile per 72-hour period:

$$wind \; gust \; footprint = \frac{(v_{max} - v_{98})}{v_{98}} * 100\% \tag{2}$$

With maximum wind gust in 72 hours at each grid point $v_{max}$, and local 98$^{th}$ percentile $v_{98}$. We use the same 72-hour periods as for the LI calculation. The corresponding cyclone tracks were derived following the tracking algorithm by Murray and Simmonds (1991) and Pinto et al. (2005). As an example, Figure 1b shows the footprint and cyclone track for windstorm Kyrill in January 2007 (Fink et al., 2009).

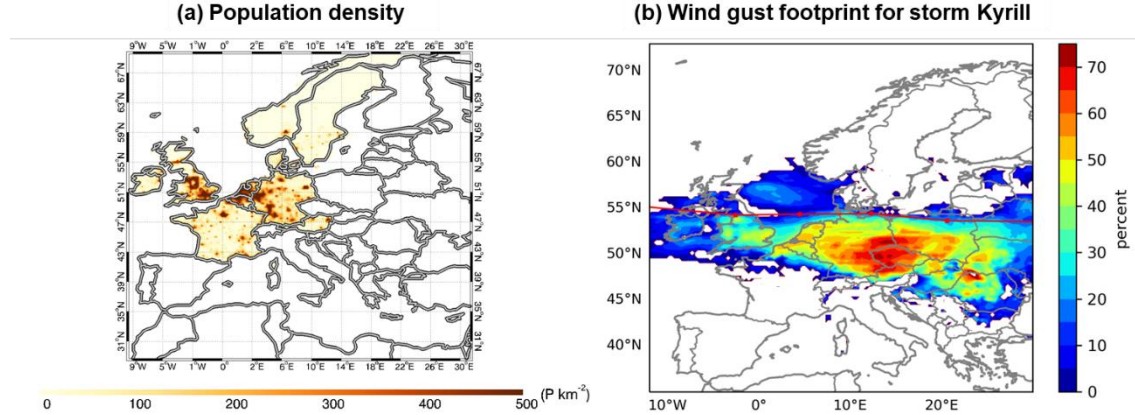

**Figure 1:** (a) Population density for 2020 derived from CIESIN for the 11 countries covered in this study. (b) Wind gust footprint for storm Kyrill in January 2007 based on ERA5. Shown is the largest exceedance (in percent) of the 98[th] percentile of daily maximum wind gust within 72 hours. The red line and dots denote the cyclone track derived from ERA5 using the tracking algorithm of Pinto et al. (2005). Please refer to Sect. 3 for detailed information on the methods.

### 3.1.3 Original and normalized loss values

For the comparison between LI ERA5 and LI ERA-Interim, we use both original loss estimates as well as normalized losses. The normalization is done with a min-max scaling approach, which scales the loss values between 0.0 and 1.0. The top storm event corresponds to the value 1.0 and the event with the lowest impact to the value 0.0. The normalized losses of all other events relate to the top event (relative ranking). We only focus on losses aggregated at country level (c.f. Sect. 3.2.1).

### 3.2 Insurance loss model

Storm loss (catastrophe) models determine the windstorm risk to residential and commercial buildings by relating wind speed to building damage (Palutikof and Skellern, 1991; Dorland et al., 1999; Gliksman et al., 2023), usually by implementing statistical modelling. Like for storm severity indices, the maximum daily wind gust speed is assumed to be the most relevant factor in these models (Dorland et al., 1999; Klawa and Ulbrich, 2003; Donat et al., 2011; Koks and Haer, 2020) and is used as the basis for the hazard component. The building damage data is usually represented using so-called loss ratios, which is the amount of insured loss occurring per district, divided by the corresponding sum of insured value (Klawa and Ulbrich, 2003; Prahl et al., 2015). For the relationship between wind and damage, also referred to as damage functions, various formulations exist in literature (see Prahl et al. (2015) for a detailed overview). These damage functions aim at describing the non-linear relation between storm intensity and actual (monetary) damage. Typically used damage functions range from exponential to power law to excess-over-threshold formulations.

### 3.2.1 Aon Impact Forecasting European Windstorm Model

In our study, we use the European Windstorm Model of Aon Impact Forecasting (Aon's IF Euro WS model), which is implemented in ELEMENTS, Aon's loss modelling platform (Aon, 2023b). The model covers 22 countries in Western, Northern and Central Europe. The aim of this catastrophe model is to provide a quantification of financial losses from windstorm risk in Europe. The model consists of three main components, namely hazard, vulnerability and exposure, plus a financial part (see Supplementary part A).

The hazard component has two parts: a historical and a stochastic event set. The historical event set comprises 26 historical storms (see Supplementary Table S1; Born et al., 2012), based on wind gust footprints built from weather station data. The stochastic event set covers 4,731 years of simulated events (Karremann et al., 2014a). The stochastic events represent physically consistent storm events and are based on outputs of the ECHAM 5 global climate model (Jungclaus et al., 2006). A combination of dynamical downscaling and statistical downscaling (Haas and Pinto, 2012) is used to produce the final high-resolution stochastic event set that is implemented in Aon's IF Euro WS model.

The exposure component typically uses a combination of Aon's client data and the PERILS industry exposure database. The component comprises five lines of business: residential, commercial, industrial, agricultural, motor and forestry (only Norway, Sweden and Finland). For our study, we only use the PERILS data for 2022 for the exposure.

The vulnerability component is divided into Chance of Loss (COL) and Conditional Mean Damage Ratio (CMDR), thereby giving a more realistic view of loss than a single mean damage ratio. The COL is applied first, rating the probability of loss for a certain wind speed and a given building. If the building is determined to have suffered a loss, then the conditional damage ratio (CMDR) is applied. The vulnerability component of the model applies IF's proprietary damage curves to calculate the physical loss for each event at each insured location. Original limits and other policy conditions are then applied per event to calculate the Gross loss and the Net loss, thus obtaining the value of insured loss in the financial component of the model.

The model is calibrated against insurance data, including PERILS data as the primary benchmark. For this reason, we assume it as a representation of a market perspective for the purpose of our paper. A more detailed description of the model is available in the Supplementary part A. For proprietary reasons, the comparison to LI is restricted to losses at country level and normalized loss values.

## 4 Comparison between ERA5 and ERA-Interim

We first analyse the sensitivity of LI to the meteorological input data. To this end, we compare ERA5 and ERA-Interim in terms of wind gust as the relevant input variable for LI. We then use both datasets to derive LI for our study domain and compare the results with respect to storm loss and storm rank.

## 4.1 Wind gust climatology

We use the 98th and 99.9th percentiles of daily maximum wind gust to compare ERA5 and ERA-Interim. The percentiles are calculated for the winter half year ONDJFM for 1979-2019, the period common to both datasets. Figure 2 shows both percentiles for ERA5 (left) and ERA-Interim (middle), as well as the absolute difference between the datasets (right). For the 98th percentile (Figure 2, upper row), both datasets show a similar spatial pattern: For most of Europe, the 98th percentile ranges between 16 and 30 m/s, with highest values over the North and Baltic Sea, and the British Isles. Except for Sweden and the Baltic region, values are in general higher for ERA5 compared to ERA-Interim. Differences reach highest values (over 4 m/s) over mountainous regions like the Alps, the Pyrenees and the Scandinavian mountains, while they are in the range of 2 m/s for Core Europe. This suggests a slight shift towards higher gust speeds in the wind gust distribution of ERA5 compared to ERA-Interim for large parts of Central Europe. For the 99.9th percentile (Figure 2, lower row), differences between the datasets are larger for all of Europe – not only in terms of magnitude but also regarding the spatial pattern. This confirms the overall shift in the wind gust distribution, but also indicates a longer tail of the wind gust distribution for ERA5 over continental Europe. Differences result most likely from the different ECMWF model version used for the reanalysis and the overall better representation of resolution and physical processes in the ERA5 setup (see Hersbach et al. (2020) for detailed information).

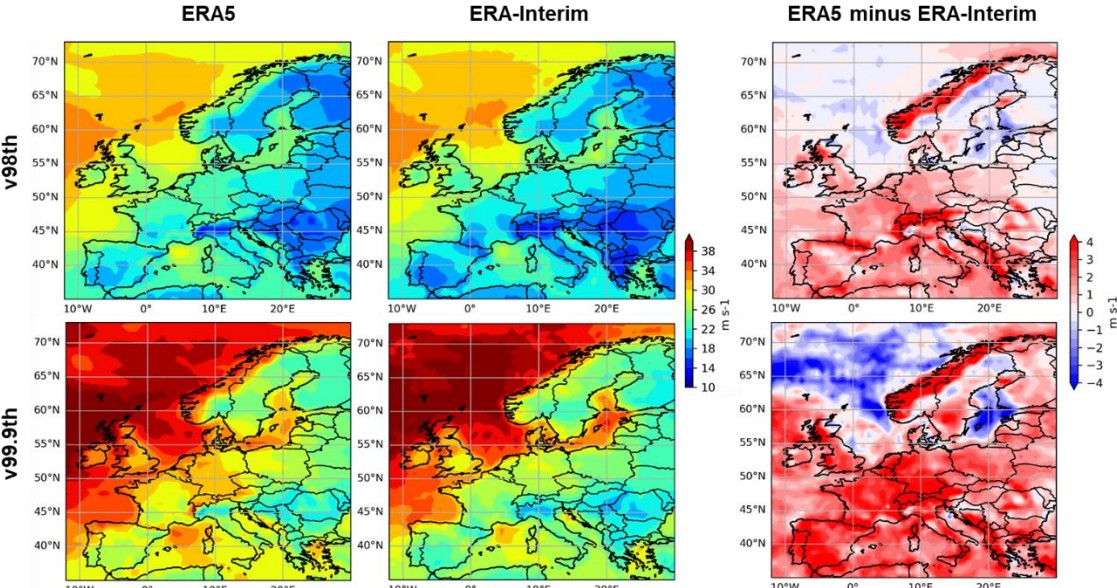

**Figure 2:** 98th percentile (upper row) and 99.9th percentile (lower row) of daily maximum wind gust for the winter half year (October – March, ONDJFM) for the period 1979-2019 derived from ERA5 (left), and ERA-Interim (middle). Difference between ERA5 minus ERA-Interim (right).

### 4.2 Storm losses and storm ranking

In the next step, we compare the loss values and the storm ranking for the 20 common most extreme storms (Top20) in the
period 1979-2019. The Top20 storms are derived separately for each country as well as for Core Europe. The storm list for
Core Europe can be found in Supplementary Tables 2 and 3. Figure 3 presents the comparison of normalized loss values
derived from LI ERA5 (x-axis) and LI ERA-Interim (y-axis) for four different regions/countries, namely Core Europe, the
United Kingdom, Germany, and France. For most events and countries, the datasets show comparable normalized losses.
Moreover, the ratio between extreme storms with high losses to extreme storms with moderate losses is similar in both datasets.
This is confirmed by the fact that most events are grouped closely around the linear regression line. Only storm Irina (October
2002) is classified as an outlier for the UK, i.e. that the difference in loss value is large based on the Inter-Quartile Range (IQR;
Dodge, 2008). The large difference between ERA5 and ERA-Interim for storm Irina can be explained by looking at the storm
footprint (Fig. S1): It is overall flatter in ERA5 compared to ERA-Interim. This is particularly the case for the UK, where the
mean wind gust over land is 12.1 m/s for ERA5 and 24.6 m/s for ERA-Interim. Therefore, the LI for storm Irina is higher in
ERA-Interim due to the cumulative effect (summation of $v/v_{98}$; see Sect. 3.1.1).

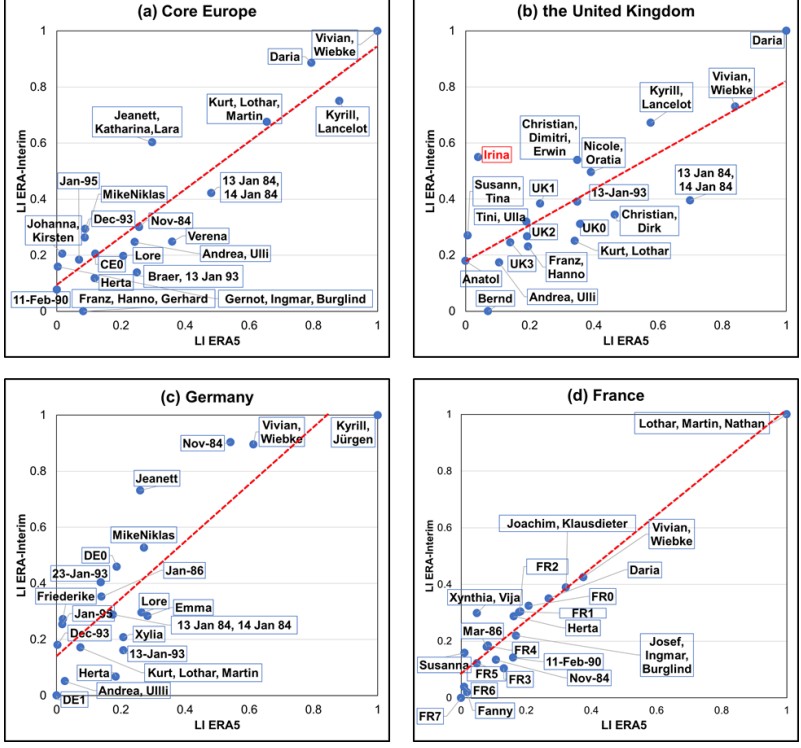

**Figure 3:** Comparison of normalized loss values based on LI ERA5 (x-axis) and LI ERA-Interim (y-axis). Depicted are the common 20
most extreme storms in the period 1979-2019 for (a) Core Europe, (b) the United Kingdom, (c) Germany, and (d) France. Corresponding
storm names to each data point are marked with a blue line. Storms without a formal name are named based on the region (e.g. CE for Core
Europe) and the loss value (starting from zero for storm with highest loss). The red dashed line denotes the linear regression line. Outlier
storms based on the IQR method (see section 4.2) are marked in red.

When comparing the original loss values (Supplementary Figure S2), the values based on ERA5 are approximately 10 times larger than those for ERA-Interim. The most obvious reason is the higher spatial resolution of ERA5 compared to ERA-Interim (roughly 3 times higher): As LI sums over all grid points with wind gusts above the 98[th] percentile, a higher number of grid

points results in an overall higher value of LI. This is confirmed by a sensitivity study, in which we re-gridded ERA5 data to the coarser ERA-Interim resolution before calculating LI (Supplementary Figure S3). After re-gridding, LI ERA5 and LI ERA-Interim are in the same order of magnitude, while the overall behaviour/order of storms does not change (cp. Figures S2 and S3). The main reason for the remaining differences between LI ERA5 and LI ERA-Interim is most likely the shift towards higher gust speeds and the longer tail in the wind gust distribution of ERA5 compared to ERA-Interim as discussed in Section

4.1.

The comparison of storm ranks between LI ERA5 and LI ERA-Interim is presented in Figure 4. Differences are generally larger than for the loss values. This is confirmed both by a higher number of outlier storms in individual countries such as France, and by an overall larger spread of events along the linear regression line.

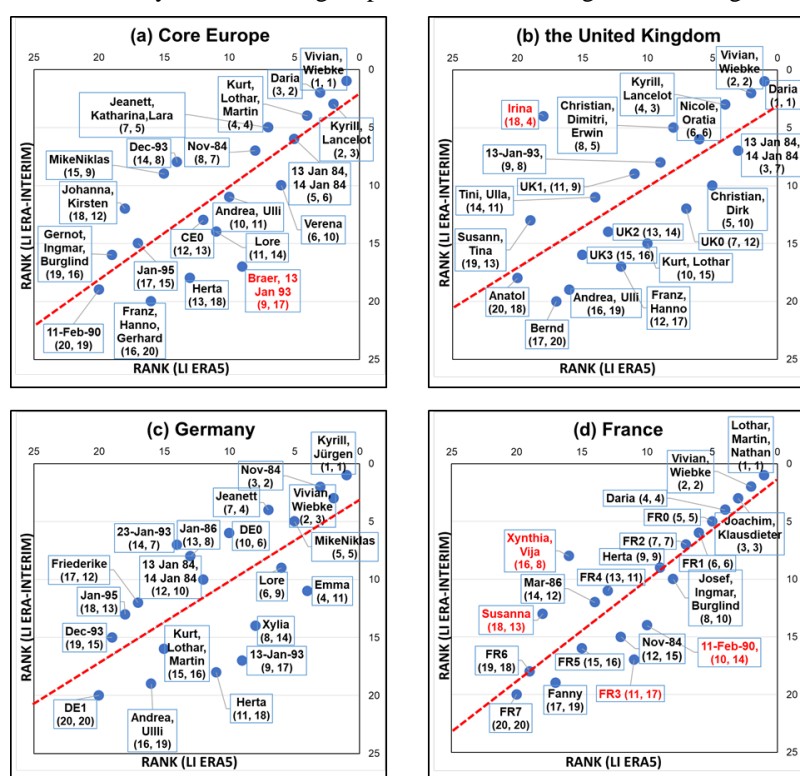

**Figure 4:** Same as Figure 3, but for the comparison of storm ranks. The values in brackets indicate the rank (first value ERA5, second value ERA-Interim).

In general, LI ERA5 and LI ERA-Interim show a good agreement. This is supported by overall high Spearman's rank correlation coefficients (Spearman, 1904; Dodge, 2008), which we computed to quantify and map the differences between the datasets across countries. In addition to the value of Spearman's rank correlation, which measures the strength and direction

of the relationship, we use the $R^2$ of Spearman's rank correlation that indicates the proportion of variance in the ranks of one variable that is predictable from the ranks of the other variable. For most countries, the correlations between LI ERA5 and LI ERA-Interim exceed 0.5, thereby confirming the good agreement between the datasets (Figure 5). Moreover, more than half of the countries have $R^2$ values above 0.40, indicating that more than 40% of the variance in the ranks of LI ERA5 is explained by the variance in the ranks of the LI ERA-Interim (Table 1). Based on these results, we focus only on LI ERA5 in the following

chapter, in order to benefit from the higher spatial and temporal resolution and the more recent data.

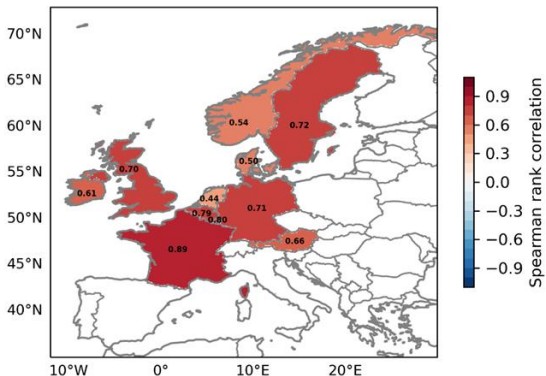

**Figure 5:** Spearman's rank correlation coefficient at country level for LI ERA5 vs LI ERA-Interim. The ranking is based on common storms per country (see Table 1 and Supplementary Figure S4).

## 5 Comparison of loss estimates from LI ERA5 and Aon's IF Euro WS model


In the second part of our study, we compare LI ERA5 to the output from Aon's IF Euro WS model, focusing on normalized losses and storm ranks at country level. The analysis is based on Aon's historical event set of insured storms in the period 1990-2020 (see Supplementary Table S1). Thus, the number of common storms between Aon's IF Euro WS model and LI ERA5 can differ in the individual countries (see Table 1 and Supplementary Figure S4). Please note that some events cannot

be clearly separated based on LI ERA5 (e.g. Lothar and Martin, see Fig. 3 and Table S2), while they are single events in Aon's IF Euro WS model. In these cases, we assign the same LI value to both storm events for the comparison between LI ERA5 and Aon's IF Euro WS model.

**Table 1:** Explained variance ($R^2$) of Spearman's rank correlation coefficient between LI ERA5 and LI ERA-Interim (2nd column), LI ERA5 and Aon's IF Euro WS model (3rd column), LI ERA5 and PERILS (4th column), and Aon's IF Euro WS model and PERILS (last column). The number of common storms per country is given in brackets.

| | LI ERA5 vs LI ERA-Interim | LI ERA5 vs Aon's IF Euro WS | LI ERA5 vs PERILS | Aon's IF Euro WS vs PERILS |
|---|---|---|---|---|
| **Core Europe** | 0.65 [20] | 0.52 [23] | 0.26 [17] | 0.57 [19] |
| **Austria** | 0.43 [20] | 0.75 [15] | 1.0 [4] | 1.0 [4] |
| **Belgium** | 0.62 [20] | 0.22 [21] | 0.09 [11] | 0.66 [11] |
| **Denmark** | 0.25 [20] | 0.41 [15] | 0.49 [5] | 0.14 [6] |
| **France** | 0.79 [20] | 0.6 [17] | 0.56 [10] | 0.54 [11] |
| **Germany** | 0.5 [20] | 0.57 [23] | 0.33 [15] | 0.47 [15] |
| **Ireland** | 0.37 [20] | 0.2 [19] | 0.49 [5] | 0.64 [5] |
| **Luxembourg** | 0.64 [20] | 0.26 [15] | 0.07 [6] | 0.43 [6] |
| **Netherlands** | 0.2 [20] | 0.64 [21] | 0.68 [11] | 0.7 [11] |
| **Norway** | 0.29 [20] | 0.4 [9] | 0.25 [3] | 1.0 [3] |
| **Sweden** | 0.51 [20] | 0.23 [13] | 1.0 [4] | 0.16 [4] |
| **United Kingdom** | 0.49 [20] | 0.36 [20] | 0.44 [13] | 0.7 [13] |

## 5.1 Case study – Storm Sabine

First, we analyse one case study in detail, namely storm Sabine that hit Europe in February 2020. We compare the normalized losses (relative ranking) and storm ranks (ordinal ranking) at country level, additionally including PERILS as a reference (Figure 6). All three datasets agree with regard to the region affected by the storm, which closely follows Sabine's cyclone track (black line and dots in left column of Figure 6). However, the normalized loss values can differ significantly in the three datasets. Values are generally higher for LI ERA5 for all countries, except for Norway, where all datasets show the same normalized loss. Aon's IF Euro WS model and PERILS show a good agreement in terms of the relative ranking of storm Sabine in the different countries. In terms of the ordinal ranking (Figure 6, lower row), Sabine is among the Top6 storms in all three datasets. However, while the ranking for Aon's IF Euro WS model and PERILS differs by no more than one position, differences are larger between LI ERA5 and Aon's IF Euro WS model/PERILS and can reach up to five positions, e.g. for the UK. In general, the agreement/disagreement between LI ERA5 on one hand and Aon's IF Euro WS model/PERILS on the other hand is different for each country and systematic differences are not apparent. Nevertheless, the results suggest that LI

ERA5 might have difficulties in clearly distinguishing individual storms from one another, i.e. that the loss values of the most extreme events are too close together. This will be examined in more detail in the following sections.

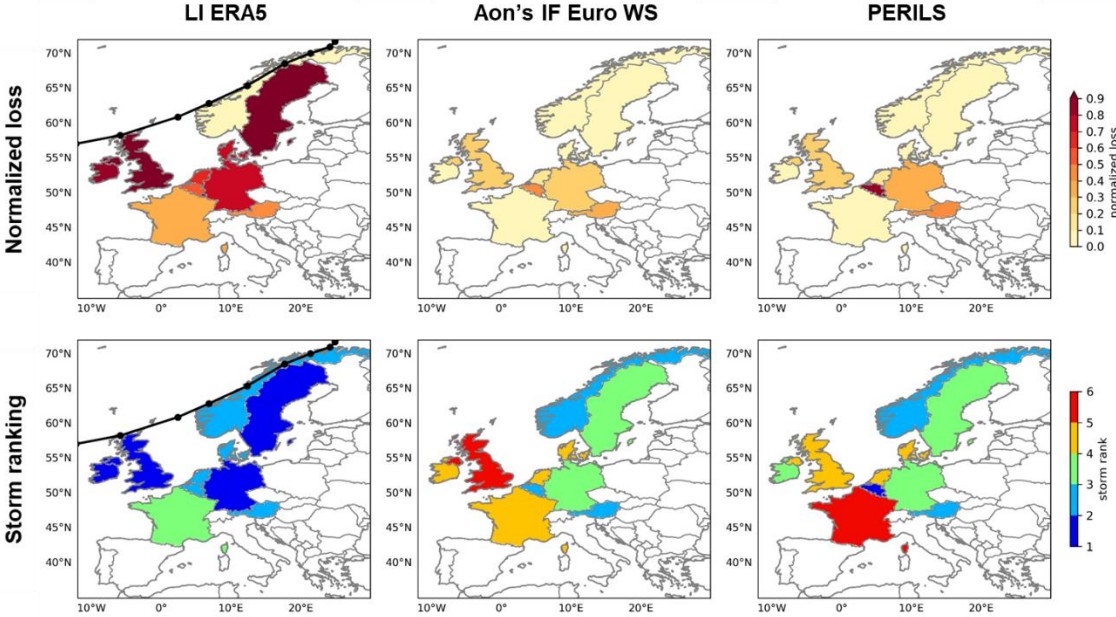

**Figure 6:** Normalized losses (upper row) and storm ranking (lower row) at country level for storm Sabine in February 2020. Losses are derived from LI ERA5 (left), Aon's IF Euro WS model (middle), and PERILS (right). The black line and dots in the left column denote the cyclone track derived from ERA5 using the tracking algorithm of Pinto et al. (2005). Losses are only shown for the 11 countries covered by Aon. The ranking is based on common storms per country (see Table 1 and Supplementary Figure S4).

## 5.2 Windstorm loss

In this section, we compare the normalized loss values derived from Aon's IF Euro WS model (x-axis) and LI ERA5 (y-axis) for all common storms for four different regions/countries: Core Europe, the United Kingdom, Germany, and France (Fig. 7).

In general, the two datasets reveal large differences. Only individual storm events like Daria in January 1990 or Kyrill in January 2007 show comparable normalized losses. This is supported by a rather large spread of storm events along the regression line (Fig. 7). Nevertheless, only a small number of storms is identified as outliers based on the IQR method – for example Sabine in Core Europe or Martin in France. For LI ERA5, the range of loss values is quite similar between larger regions like Core Europe and smaller regions (individual countries). Aon's IF Euro WS model, on the other hand, reveals a

different range of loss values for different regions. Within individual regions, Aon's IF Euro WS model shows a clear distinction between extreme "high loss" storm events such as Daria and those events with "moderate" losses (e.g. Isaias). Normalized loss values between those events can differ by a factor of up to 1000 for single countries. This distinction is less pronounced in LI ERA5 (see e.g. Fig. 7b), where the individual storm events are closer together and usually differ by a factor of less than 100 in terms of their respective normalized loss. Such differences are not uncommon when comparing loss datasets

(Moemken et al., 2024).

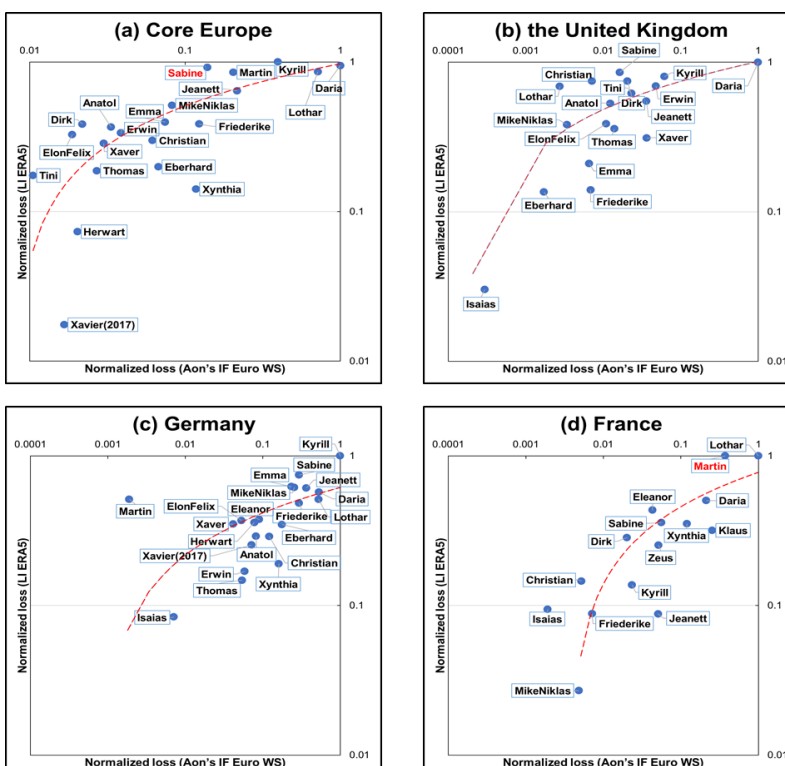

**Figure 7:** Comparison of normalized loss values between Aon's IF Euro WS model (x-axis) and LI ERA5 (y-axis). Depicted are the common most extreme storms for the period 1990-2020 for (a) Core Europe, (b) the United Kingdom, (c) Germany, and (d) France. A logarithmic scale is used for the axes. The red dashed line denotes the logarithmic regression. Outlier storms based on the IQR method are marked in red. Please note the different scales.

In a sensitivity study, we tested whether the differences between LI ERA5 and Aon's IF Euro WS model result from the different event definition – 72-hour periods vs 24-hour periods. With this aim, we calculated LI ERA5 for running 24-hour windows. The comparison of normalized loss values is shown in Supplementary Figure S5 (see Figure S6 for storm ranks). Overall, we find no systematic reduction in the differences between LI ERA5 and Aon's IF Euro WS model when using 24-hour windows instead of 72-hour windows. For some storms and/or countries, differences decrease with a shorter event definition (e.g. for Germany), while for others they increase (e.g. Core Europe). Moreover, the number of common storm events decreases with a shorter event definition for LI ERA5 (not shown).

### 5.3 Storm ranking

We also compare LI ERA5 and Aon's IF Euro WS model in terms of storm ranks for the common most extreme storms per country. Figure 8 shows this comparison for Core Europe, the UK, Germany and France. As for the normalized losses, we see rather large differences between the datasets, though less pronounced. Most events show rank differences in the range of zero to three positions. Only in the case of individual storms, such as Klaus in Core Europe or Martin in Germany, can rank differences reach up to 16 positions. These events are also marked as outliers based on the IQR method.

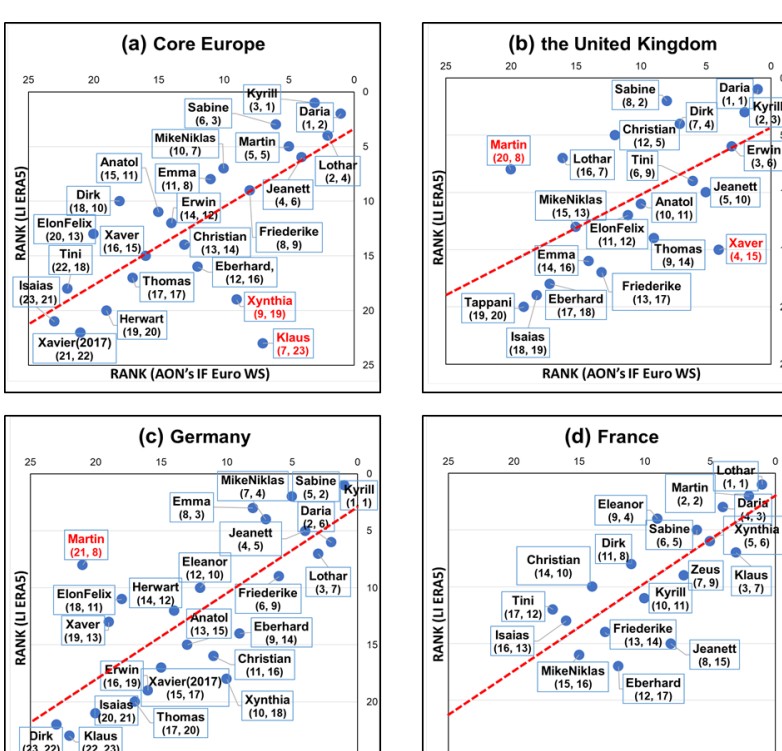

**Figure 8:** Same as Figure 7, but for the comparison of storm ranks. The values in brackets indicate the rank (first value Aon's model, second value ERA5).

Finally, we compare Spearman's rank correlation coefficients (Figure 9) and the corresponding explained variance ($R^2$; Table 1), again including PERILS as a reference. Figure 9 displays the correlation coefficients for each country, providing a clear depiction on the agreement or disagreement between LI ERA5, Aon's IF Euro WS model and PERILS. For most parts of Central Europe, LI ERA5 and Aon's IF Euro WS model show a high agreement, with correlation values reaching up to 0.86 for Austria. Lower correlations with values below 0.5 and therefore larger differences can be found for Ireland, Belgium and Sweden. The correlation pattern between LI ERA5 and PERILS looks similar, with overall lower values. Only the perfect anti-correlation for Sweden and the perfect correlation for Austria are striking. However, these values could be due to the small sample of common storms (see Supplementary Figure S4) and should therefore be viewed with caution. The comparison of Aon's IF Euro WS model and PERILS reveals mostly high correlation coefficients, ranging between 0.69 for Germany and 1.0 for Austria and Norway. In terms of the explained variance, Austria exhibits the highest $R^2$ value when comparing LI ERA5 against Aon's IF Euro WS model and PERILS (Table 1). This result suggests that for Austria, over 70% of the variation in the ranks of loss from one dataset can be explained by the variation in the ranks of the other loss values. Due to the small sample of common events in some countries, some correlation values in this comparison should also be treated with care.

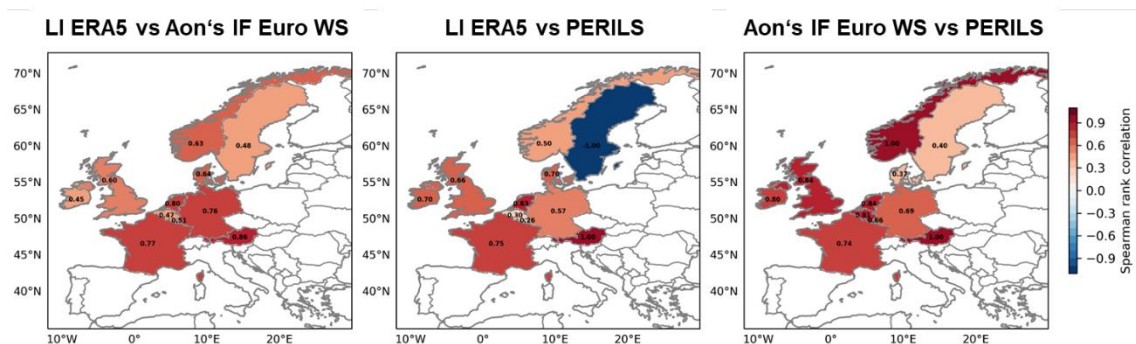

**Figure 9:** Spearman's rank correlation coefficient at country level for LI ERA 5 vs Aon's IF Euro WS model (left), LI ERA5 vs PERILS (middle), and Aon's IF Euro WS model vs PERILS (right). The ranking is based on common storms per country (see Table 1 and Supplementary Figure S4).

## 6 Summary and discussion

In this study, we compared estimated windstorm losses over Europe from the meteorological loss index LI and the catastrophe windstorm model of Aon Impact Forecasting, used in insurance. Furthermore, we tested the sensitivity of LI to the meteorological input data by using both ERA5 and its predecessor ERA-Interim. The main results can be summarized as follows:

- For all of Europe, LI values are higher for ERA5 than for ERA-Interim (by roughly a factor of 10). The main reason is the higher spatial resolution in ERA5. Additionally, the wind gust distribution in ERA5 is slightly shifted towards higher values and has a longer tail. With regard to normalized losses and storm ranks, LI ERA5 and LI ERA-Interim show a comparable behaviour for Core Europe with Spearman's rank correlation mostly ranging between 0.61 (Ireland) and 0.89 (France).

- Compared to Aon's IF Euro WS model, LI ERA5 shows overall lower normalized loss values, while the storm ranks are comparable for most of Core Europe (correlations between 0.45 and 0.8). Moreover, Aon's IF Euro WS model reveals a clearer distinction between high and moderate impact events. The difference between the highest and lowest insured loss, as given by Aon's IF Euro WS model (e.g. Daria vs Isaias in UK, see Fig. 7b) is 3 orders of magnitude, while the corresponding LI ERA-5 difference is typically 1 to 1.5 orders of magnitude. In addition, the catastrophe model shows a clear regional dependency of loss values. This regional dependence is less pronounced in LI ERA5.

In previous studies, LI has been calculated and analysed for a variety of reanalysis datasets with different spatial and temporal resolutions: ERA-40 with 1.125° and 6-hourly resolution in Pinto et al. (2012), NCEP with 1.875° and 6-hourly resolution in Karremann et al. (2014a), or ERA-Interim with 0.75° and 6-hourly resolution in Priestley et al. (2018). In line with our results, these studies show that the magnitude of LI is sensitive to the spatial resolution of the underlying dataset. Nevertheless, they all agree on the general (regional) behaviour of LI. Another reason for the different LI values for ERA5 compared to ERA-Interim is a slight shift towards higher gust speeds and a longer tail in the wind gust distribution of ERA5. This is in line with

Minola et al. (2020), who compared wind gust data from ERA-Interim and ERA5 with observational data across Sweden. They find an overall better agreement between observations and ERA5, although some discrepancies persist in regions with complex topography. We therefore conclude that it is adequate to use the recent ERA5 dataset for the comparison to the insurance model in the second part of our study.

One reason for the differences between the meteorological index and the catastrophe model of Aon Impact Forecasting is their different methodological design: First, Aon's IF Euro WS model uses a 1-day window for the loss calculation, while LI ERA5 is based on 72-hour windows. Thus, Aon's IF Euro WS model is better able to separate storm events in short succession (like Lothar and Martin in December 1999). In a sensitivity study, we could show that using a 24-hour event definition for LI ERA5 does not lead to a systematic reduction in the differences between LI ERA5 and Aon's IF Euro WS model (Sect. 5.2).

Therefore, we decided to stick to the 72-hour event definition in LI ERA5. This has several advantages: We are able to capture the entire windstorm footprint (Hewson and Neu, 2015). Additionally, the 72-hour event definition corresponds to a definition often used in reinsurance treaties (the so-called 72-hour-clause; Klawa and Ulbrich, 2003; Karremann et al., 2014a). Finally, the correlations between LI ERA5 and Aon's IF Euro WS model are higher when using 72-hour windows, especially for Core Europe. Another methodological difference is the consideration of different risk components. LI ERA5 only includes the

hazard component and an estimate for the exposure component, while Aon's IF Euro WS model additionally includes a sophisticated engineering-based vulnerability component that takes e.g. building resistance, loss frequency due to quasi-random effects and local societal adaptations into account.

Aside from that, our study reveals some shortcomings of the two approaches. As all meteorological indices, LI relies upon the quality of both the underlying wind data and the impact function used for the calculation of loss. In the specific case of LI, the

390 initial index was developed and evaluated for Germany by Klawa and Ulbrich (2003), employing insurance data of Munich Re and GDV ("Gesamtverband der Deutschen Versicherer e.V."). In a follow-up study, Karremann et al. (2014b) were able to demonstrate that the chosen 98[th] percentile is an appropriate threshold to identify extreme storm events over Central and Western Europe. Nevertheless, they also point out that the 98[th] percentile might be too low for South Eastern Europe, the Mediterranean and Scandinavia. For these regions, Karremann et al. (2014b) suggest the use of a fixed, reasonable threshold

below which losses are improbable. Moreover, the usage of present-day population density as proxy for exposure levels might lead to an overestimation of loss values (Koks and Haer, 2020). Furthermore, LI depends on the used gust data. The ERA5 wind gust data, we use here, is based on the parameterization approach by Panofsky et al. (1977). While this approach performs well in flat terrain, it is sensitive to the local parameterization of the roughness length (Born et al., 2012; van den Brink, 2019). Finally, the LI index is missing a detailed damage component. The applied cubic relation tries to mimic the non-linear response

of buildings to wind gusts. However, compared to the market perspective of Aon's IF Euro WS model, LI ERA5 seems to struggle with capturing this non-linearity, especially for the high impact events at the tail of the gust spectrum. In some extreme cases, certain exposures (e.g. greenhouses, timber building or agricultural buildings) may have vulnerability functions approximating a step-function. Various studies tested different formulations of meteorological indices, also considering different exponents (e.g. Klawa and Ulbrich, 2003; Pinto et al., 2012; Prahl et al., 2015; Gliksman et al., 2023). All these

studies agree that the performance of the different indices depend on the underlying event set. For some events, formulations with higher exponents seem to estimate windstorm losses better, while for other events, the cubic relationship provides results that are more realistic. In this sense, no formulation clearly outperforms the others. Aon's IF Euro WS model on the other hand, includes no information on not-insured market loss. Additionally, insurance data in general depend on the insurance coverage and policy in single countries. Both factors might result in an overrepresentation of windstorms that hit countries with high market coverage (Moemken et al., 2024).

The current study is, to our knowledge, the first to compare a full insurance windstorm model (which is not publicly available) to a simplified meteorological loss index. For this reason, and due to some proprietary restrictions, we decided to focus on a straightforward comparison of the two methods. Overall, our results suggest that the loss distribution in LI is not steep enough and accordingly the tail is too short, leading to an underestimation of high impact windstorms compared to the market perspective derived from the insurance catastrophe model. Nonetheless, LI is an effective index precisely because of its simplicity since it only considers wind gust and population density. Although it cannot be used to price a storm (due to the missing vulnerability information), it is suitable for estimating the impacts and rank events. The first comparison between a meteorological index and a full commercial windstorm model could serve as a reference for future studies focussing on the development and improvement of both storm loss models and storm severity indices.

## Data availability

The ERA5 and ERA-Interim reanalysis data (input for the loss index) can be downloaded from the Copernicus Climate Change Service (C3S) Climate Data Store (https://cds.climate.copernicus.eu/). Access to PERILS is granted via an annual subscription in accordance with a PERILS database license.

## Author contribution

JM, AMR and JGP conceived and designed the study. IA performed the data analysis and made the figures with the help of AG, AB and LB. JM wrote the initial paper draft. All authors discussed the results and contributed with manuscript revisions.

## Competing interests

One of the authors (JGP) is member of the editorial board of Natural Hazards and Earth System Sciences.

## Acknowledgements

IA was funded by European Union's Horizon 2020 research and innovation programme under Marie Skłodowska-Curie grant no. 956396 (EDIPI project). JM was funded by the Bundesministerium für Bildung und Forschung (BMBF; German Ministry

for Education and Research) under project "RegIKlim-NUKLEUS" (01LR2002B and 01LR2002B1). JGP thanks the AXA Research Fund for support. AMR was supported by the Helmholtz 'Changing Earth' program. We thank the German Climate Computing Center (DKRZ, Hamburg) for computing and storage resources. IA thanks Petr Svoboda and the team of Aon Impact Forecasting for support during her stay in Prague. We thank Ting-Chen Chen for preparing the cyclone track data, Jisesh Sethunadh for discussions about the LI calculations, and Federico Stainoh (all KIT) for suggestions on the statistical analysis methods. We thank the three reviewers and the editor for their valuable comments.

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
