# Peer review of "Insurance loss model vs meteorological loss index – How comparable are their loss estimates for European windstorms?"

_Natural Hazards and Earth System Sciences, 2024_

## Referee Comment (RC1)

**Review of "Insurance loss model vs meteorological loss index – How comparable are their loss estimates for European windstorms?" by Julia Moemken, Inovasita Alifdini, Alexandre M. Ramos, Alexandros Georgiadis, Aidan Brocklehurst, Lukas Braun and Joaquim G. Pinto**

The authors describe a comparison between reanalysis-based Loss Index with losses obtained from the European Windstorm Model of Aon Impact Forecasting. In addition, two different reanalysis products (of diferent generations) are compared to assess the impact of the reanalysis on the actual Loss Index. The study addresses a very interesting subject and the ability of the authors to compare reanalysis-based Loss Index with the Aon Impact Forecasting model is an opportunity and very relevant for the community. However, I found the manuscript a bit difficult to follow at times but perhaps more importantly, I am concerned about the representation of the wind gusts that is used in the manuscript. The authors are right with their statement on line 137 that "the maximum daily wind gust speed is assumed to be the most relevant factor in these models". My most serious concern relates to this. A bit more detail is provided below. There are a few other issues the authors may want to look into.

My advice to the editor is to accept with major revisions.

*Major concern:*

line 105-110. There is a bit of concern on the parameterization of wind gusts from the reanalysis data. One approach is to use the local near-surface wind speed and its standard deviation in order to estimate the gust (like the Panofsky et al 1977 approach used in the manuscript). This approach makes use of similarity theory, and relates the gust to the friction velocity. The approach performs well in flat terrain, but is sensitive to the parametrization of the local roughness length. The accuracy of the estimated gusts relies heavily on the roughness map that is used, especially when the resolution of the NWP model increases and detailed information about the land-use (and the associated roughness lengths) is required. Errors in the supplied roughness lengths will directly influence the calculated gusts, which is a disadvantage of this approach.

My suggestion is to add a brief analysis where the Panofsky et al (1977) approach is compared to an alternative approach which is specifically suitable for use in a reanalysis product (van den Brink 2019). It links the 1-hour wind speeds at height (which is a standard output of the reanalysis) to 10m wind gusts. This comparison can be done over the entire domain or for specific storms.

To complete this analysis, actual observations of wind gusts should be combined to this assessment. Wind gust values for Europe can be obtained from the European Climate Assessment & Dataset at www.ecad.eu. If you have troubles finding the right data, simply contact ECAD staff.

This additional analysis assess the quality of the parameterization used for the wind gust calculations which is central to this study. The quality of the parameterization is therefore essential and requires a bit more scrutiny that the brief comments that is currently found in the manuscript.

*Other aspects the authors may want to look at:*

- section 3.1: fig. 2c shows that ERA5 generally has higher wind gust values than ERA-Interim, but the strongest differences are found over area with complex topography, like the Pyrennees, Alps, Norwegian coast and (perhaps) the Scottish highlands. The higher values over Europe are likely related to a mix of better physics and higher spatial resolution - as the authors correctly state. Could you explicitly state the spatial resolutions of ERA5 and ERA-Interim in the section where they are introduced? Now the resolution of ERA-Interim is mentioned on line 308 in the very last section of the manuscript. With a coarser resolution, complex topography will be much less well represented and peaks and vallayes will be less pronounced which directly affects the wind gust. Perhaps good to make this explicit in the discussion of the reanalysis.

- line 204-206. I am afraid that I fail to see why storm Irina is such an outlier. The Loss Index for ERA-Interim is for this storm much larger than for ERA5, but the storm footprint (fig. S1) does not really show a much larger region where the footrpint != 0 over the UK. ERA5 does show higher values (mostly because of the high resolution of ERA5 I guess). So, what do I fail to see in the explanation?

- Figure 3 and 4: it would be interesting to add an analysis where ERA5 is first regridded to the ERA-Interim resolution, and then the LI diagrams are made. This analysis gives a clue if it is the improved physics in ERA5 that makes the difference or that the increase in spatial resolution makes the difference. This would be nice to add to the Supplementary material. This analysis could then provide the basis for Section 5, bullet 1: I have not seen evidence that it is the resolution that makes ERA5 better than ERA-Interim (although this is likely).

- Section 5: bullet 1: the distribution of wind gusts may be shifted right in ERA5, but the footprint uses the 98th percentile - which is also shifted right. So this argument does not make sense.

- Figure S1: In the caption of the figure you write "Shown is the percentage of the maximum wind gust in 72 hours that exceed the 98th percentile of daily maximum wind gust." If you aim to show the outcome of equation 2, the this should be something like "Shown is the strength of the maximum wind gust in 72 hours as deviation from the 98th percentile and normalized with the 98th percentile."

**REFERENCE**
van den Brink, H. W. (2019). An effective parametrization of gust profiles during severe wind conditions. Environmental Research Communications, 2(1), 011001. DOI 10.1088/2515-7620/ab5777

**NUTRevision for paper: nhess-2024-16: Insurance loss model vs meteorological loss index – How comparable are their loss estimates for European windstorms**

Dear authors, I read your paper with curiosity. I think it is interesting for scientists to begin collaborating with the insurance sector, so that researchers and insurers can better understand the impacts of severe weather on different socio-economic sectors. I do think it is a timely topic, but I believe the manuscript requires quite some work to become clear and deliver its core messages effectively. Hence, my recommendation is a *Major Revision*, and I hope the comments in this document will be helpful. Good luck.

**Major comments**

In this section you can find comments in two categories: structural and data analysis. For the structure, I have the impression the paper could benefit from a clearer structure, with a better division between data descriptions and the methods, whereas for the analysis comments there are parts that remain unclear.

**1) Structural comments**

**Introduction:** The introduction requires some streamlining, since it intertwines motivating reasons to carry out such a study with lengthy descriptions of previous work. As a result, it is difficult to follow the storyline the authors wish to convey. For example, in L36-L51 you begin talking about the hazard, exposure, vulnerability framework, but this somehow becomes diluted in the rest of the paragraph. It might be helpful for readers to center the introduction about these three components of risk management using the risk propeller figure, so that the references to these multiple insurance companies and other articles are somehow anchored to this image. Then in L65-L68 the authors roughly describe the analysis that will be doing, which I find too detailed for an introduction, to then explain the paper structure, which jumps back to the general scope. Overall, I think this section requires streamlining and making sure the message the authors wish to convey is effectively delivered.

**Data and methods:** I would recommend re-structuring this section. While reading, there are parts mixing data description with the methods, which interrupt the flow. For example, L102-L114 describe the ERA5 data (and other generalities) right after the equations for LI are presented. Then in L116 the flow is recovered. Same goes for the description of PERILS in L154-L163. On the one hand, in the introduction the authors mention a hazard-exposure-vulnerability schema. On the other hand, I have the impression that the hazard

component is ERA5, the exposure is PERILS, and the vulnerability the data/curves from AON. So I would recommend restructuring this section in 2.1.1 - Hazard; 2.1.2 - Exposure; 2.1.3 - Vulnerability and then a 2.2 - Meteorological loss index and 2.3 - Catastrophe model that are thoroughly explained without data description intrusions.

**Summary and discussion:** I find this section long and I am not sure what the main conclusions of this work are. Is there any way of separating the "more technical" discussion part from the "more abstract" conclusions? Overall, I do not see the "take home message", or how does this relate with the two very concrete research questions posed in L60-L64. Also, how might the insurance sector be using the insights gained in this study?

**2) Data analysis comments**

**2.1 - Meteorological loss index**

In L86 the text say "Losses are proportional to the wind power or the wind kinetic energy flux....". Perhaps softening or extending this description might be useful for a generic reader to comprehend the meaning and implications of this.

In L87-L88 you mention that "...only the 2% of wind gusts....cause damage". I am missing here some elaboration about what are the damages that you have in mind. Are we talking infrastructural damage? Agricultural damage? To public or private assets? Is personal propriety included here? If this is one of the four assumptions in the paper, I would expect to have a solid description of what is the meaning of "damage" for the authors in this work.

In L90 you mention "In the case that no insurance data, population density can be used as a proxy for the exposure component". Indeed, but then does it mean that you are focused in damage in cities, hence, roads, agriculture, or forestry damages are out of the study? Also, how frequently do you bump into records that have no insurance data associated? I think this study could benefit from some extra clarity on how much insurance data is available, as long as its contents. This might help at assessing whether population density is a matching candidate for the insurance data or requires combining it with other layers (e.g. land use, urban tree, urban morphology).

Also, I wonder how the different spatial dimensions are accommodated in this analysis. For example, population density from CIESIN at 0.25deg is roughly 30km, but then how insurance data are aggregated? Per country? Per NUTS region? And how does this relate with the spatial resolution of ERA5, ERA5-Interim and the catastrophe model from Aon? I

believe it would be useful to have a section discussing the harmonization of the spatial dimension, so that it is clearer what the two models receive as input.

**4.1 - Windstorm loss**

Here in Figure 5 some results are visualized in the geographic space. In this figure I have two comments. First, the results are presented in a per-country basis, but the analysis seems to have been carried out on pixels much smaller than the country surface. I wonder if results can be presented using NUTS 2 regions or a spatial unit that is closer to the spatial dimension of the analysis. If results are aggregated for the sake of visualization, this would be understandable, but then I would expect a clearer description of the treatment of the spatial dimension throughout the manuscript. What is the resolution of the insurance data? How are all these harmonized? Second, the colorscale chosen in this figure might not be ideal to visually perceive differences. Perhaps a sequential colormap (with 3 colors) or a perceptually uniform sequential colormap (eg. Like viridis) might be a better choice to guide the reader to the differences you describe.

Also, I do not really understand how to interpret the Figures with the storm ranks. What helps the reader understand what is relevant?

**Minor comments**

- L43-L51: I think this might be a bit too detailed for an introduction, perhaps I would recommend streamlining this part.
- L44: "a direct view on the impacts" -> what type of impacts? Economic? Human?
- L50: "Actual loss reports... are usually not publicly available". Is this a limitation for this study, given the access to Aon data?
- L62: "In our study": I thought it was making reference to (Moemken et al., 2023) that is previously explained, so perhaps "in this study" reconnects with the paper at hand.
- L84: "Loss Index (LI)" has been previously defined
- L84: "...in the adaptation of Karremann et al (2014)" this line is written just as at the end of the introduction
- L131: The red line of the figure is not too visible on printed paper
- L168: Is this the right title for a section also containing storm rankings? Is there a more informative way of conveying the content of the section?

- L175: In Figure 2, I believe panel (c) is visually cluttered and it is hard to interpret the map with the overlying grid. Please, consider using something as "bivariate color scales" to represent 3 dimensions in a 2D space.
- L177: I understand that you have to cut somewhere, hence the choice of the 98th percentile, but I am lacking some extra motivation on why this is the right choice for this analysis. Can you elaborate? Is it common practice? What happens if you pick the 97th percentile?
- L178: "...is calculated for the winter half year". I think it would be good to make explicit the reason of this choice
- L184: I am not familiar with the definition of "Core Europe", but I would suggest coming up with a more neutral definition for a portion of the European continent. Is Central Europe and British Isles (CEBI) a reasonable option? Or study region/area? Is it correct to call it "Core Europe" after Brexit? I think it is important to strive for neutral terminology as much as possible.
- L205: "...Irina is easily...": non-neutral language
- L209: "...on a paired Wilcoxon Signed-Rank test". I am acquainted with Pearson, Spearman and Kendall indices (which I perceive as more common), hence I believe the choice of Wilcoxon could be better justified. What do you expect to obtain from its application? What are the benefits?
- L225: In Figure 4, I can see the names with a pair of integers in brackets. What are these? Are they part of the Wilcoxon test? If this is relevant information, I believe it should be elaborated somewhere. In the end, this also suffers from visual cluttering, so I think it is important to guide the reader on how they should interpret these plots.
- L226: Figure captions should be self-explanatory, so in my view using "Same as Figure X" is not an appropriate practice. Other figures also have this problem.
- L270: In the figure, is it okay that the axes bounds are not the same?
- L292: "...the wind gust distribution in ERA5 is slightly shifted towards higher values". I can find this explanation in several places along the document, but then my comment would be, why don't you show a couple of histograms illustrating the distribution of wind gust in your study period? It can go to the supplementary material if it does not belong here.
- L304: By inspecting Figure 8 (and its description) I am interpreting that the plain LI ERA5 index does a good job, and the improvements provided by Aon's model might not be substantial. Perhaps if the analysis data was aggregated at a NUTS 2 region (provinces/subnational regions), more differences would pop up.
- L328: I found the explanation of the 98th percentile. Perhaps this could be moved up to the Data & Methods section.

- L340: Another mention to the tail of the distribution. I recommend adding this to the manuscript, since this seems to be a relevant piece of the discourse.

---

## Referee Comment (RC2)

**Insurance loss model vs meteorological loss index – How comparable are their loss estimates for European windstorms?**

by

Julia Moemken, Inovasita Alifdini, Alexandre M. Ramos, Alexandros Georgiadis, Aidan Brocklehurst, Lukas Braun, Joaquim G. Pinto

The paper compares estimated windstorm losses using the meteorological Loss Index (LI) with losses obtained from the European Windstorm Model of Aon Impact Forecasting. The paper has a twofold focus. First, how sensitive the loss estimates of the meteorological Loss Index (LI) is to the meteorological input data using two different reanalysis products. Secondly, comparison of windstorm loss estimates from the meteorological Loss Index estimates and an insurance loss model using the Aon's Impact Forecasting model. They conclude that the loss distribution in the LI estimate is not steep enough and accordingly the tail of the loss index distribution is too short, leading to an underestimation of high impact windstorms compared to the insurance catastrophe model. Despite of its differences, compared to the Aon model, the authors conclude that the meteorological Loss Index model is suitable for estimating the impacts and rank events.

**General comment**

It is clearly within the scope of NHESSS and is written in a clear and well-structured way. The research questions are clearly outlined in the final part of the introduction.  The paper has the potential to shed light on the differences between a rather simple, but well documented open access approach and a more refined proprietary commercial product. This is very welcome contribution that could inform the community on the differences between approaches pursued by the academic community and private sector.

I find the analysis to be somewhat superficial and with few exceptions it consists of correlation analysis and scatter plots. The correlation analysis is rather hard to interpret and to properly answer the research question "How comparable are windstorm loss estimates from this meteorological index and an insurance loss model?" a much more multi-faceted approach is needed.

Aon's Impact Forecasting model is a commercial windstorm model and is treated as a black box. From a scientific point of view this seriously hampers both the depth of the analysis and the information value of the conclusions. Questions like why the two approaches differ can only be answered with statements concerning the shape of the final loss distribution and there is no comparison against reality which prevents any statements on quality.

I believe the manuscript requires major work before it can be accepted and recommend a resubmission or a major revision.

**Specific comments**

- Abstract: Research question two (comparison between the models) which from the title of the paper is the most important only have three sentences in the abstract.

- Abstract: It would be beneficial if the qualitative statement such as "shows comparable storm ranks", "yet it is an effective index", "higher LI values for ERA5 than for ERA-Interim" are justified by some numbers.

- Introduction: The introduction starts out with a rather weak statement "In Central and Western Europe, windstorms are among the major natural hazards". I am sure a more precise description of the importance can be found. For example in https://wmo.int/publication-series/atlas-of-mortality-and-economic-losses-from-weather-climate-and-water-related-hazards-1970-2021

- Introduction: There is a rather lengthy paragraph on loss datasets, but very little on different wind-damage functions. Given that the paper is on estimation of loss and particularly the steepness of the loss with increasing winds, I suggest that more emphasis is put on the describing the literature on wind-damage relations and less on the damage datasets (which was reviewed in the authors previous paper).

- Section 2:1: The LI equation is summing up grid squares without any grid square area scaling. A storm centered at 45N will have grid squares that are around 40% larger than one centered at 60N. Thus, equally sized storms will have rather different number of grid squares, making the LI values not directly comparable. This should be mentioned in the text as a possible weakness of the LI when used over regions spanning a large area in the north-south direction.

- Section 2:1: Only events with LI values above a certain threshold are kept. The authors note that they are only interested in extreme storm events and they use a threshold "which corresponds to the selection of an average of five events per season". Does this mean that if you have 41 years of data there will be 41*5=205 events (assuming that there is only one season in this study, ONDJFM)?

- Section 2.1: The post-processing of wind gusts based hourly data is difficult and needs a more detailed description. I wonder if this step is really necessary? Post-processed wind gusts based on hourly data tend to be strongly correlated to the hourly winds and have similar distribution shape. If this is the case the $(v/v\_98)^3$ ratio in the Loss Index (LI) should be almost the same if the daily max hourly wind is used instead of the wind gust data. If this is the case, there is no need to introduce a questionable wind gust parameterization, the $(v/v\_98)^3$ ratio could be calculated directly from the daily max hourly wind speed.

- Section 2.1: Make a separate section for the input data/information. (ERA, population density data, storm names etc.). It should not be described in the Meteorological loss index section.

- Section 2.2: The first paragraph is not about the Aon model or the Perils data. Consider removing it.

- Section 2.2: The Aon model is not described in much detail. Is there no more information available on the model? The paper would benefit strongly from a more detailed description of the Aon model.

- Section 2.2: Consider making a separate data section where the PERILS data can be described.

- Section 3.1: Section 3 starts with focusing is on the comparison between ERA5 and ERA-Interim 98th percentile wind gust. The 98th percentile is the loss-no loss threshold which is only marginally interesting. As the loss is increasing with $(v/v\_98)^3$ a more revealing analysis would be the difference between for example $(v\_99.7/v\_98)^3$ (99.7 percentile is the once-a-year value) or $(v\_max/v\_98)^3$. I think this would better describe how the differences between ERA5 and ERA-Interim would affect the loss calculations.

- Section 3.2: It is rather unfortunate that the losses is max-min normalized when ERA5 and ERA-Interim are compared. Is it not more revealing to do a grid square area scaling to get rid of the grid square dependence in the LI method and then look at the remaining differences in the Loss Index distributions instead of showing the rescaled versions?

- Section 3.2: The storm losses and storm ranking comparison is done using Pearson and rank correlations. For the Pearson correlation to be informative the data need to be normally distributed. It is pretty clear that the loss data is heavily skewed and the Pearson correlation becomes pretty meaningless or at least very hard to interpret. The $R^2$ will not be the variance the two datasets have in common. Either the data must be transformed to become normally distributed (Box-Cox or similar) or other measures of similarity should be applied. It also seems that the authors have used the paired Wilcoxon Signed-Rank Test on the Pearson correlations. But as the Pearson correlation assumes normal distributed data, the t-test would be the appropriate test.

- Section 3.2: The authors note that they are "ranking for the 20 common most extreme storms", but it is not necessarily the most extreme storms, but the one with largest loss values.

- Section 3.2: How is the "20 common most extreme storms" found and how will this affect the ranking analysis? If a common storm is no. 18 in one dataset and no. 24 in another, but picked as one of the 20 common most extreme storms, will they be reranked (between 1 and 20) and the ranking analysis based on the reranked values?

If they are based on reranked values, the results may not be representative for the original datasets.

- Section 3.2: The authors state that the LI values based on ERA5 are approximately 10 times larger than for ERA-Interim due to smaller grid squares. To highlight possible other reasons, the LI values could be scaled with grid square area, to investigate if any systematic differences that was not due to the obvious grid square size dependence.

- Section 3.2: Storm loss rankings are based on shared variance of the ranked scores (rank correlation squared). It is not mentioned in the text, but I guess this is based on the Spearman rank correlation?

- Section 3.2: One might wonder if the correlation analysis really is the way is to go for showing differences in the loss estimates? I have a hard time understanding what a $R^2$ of 0.5 really means. According to the paired Wilcoxon Signed-Rank Test correlations squared down to 0.27 and rank correlations squared down to 0.18 does not indicate significant differences between LI ERA5 and LI ERA-Interim. Given the low number of events and possibly a few events that are very different in the two sets, there might be other methods than correlation analysis that are more useful and easier to understand. Maybe categorizing the losses, Kendal tau, Rank Biased Overlap, Goodman and Kruskal's gamma, precision etc. or other methods can be considered.

- Section 4.1: The same comments on the use of Pearson correlation etc. mentioned in section 3.2 applies on this analysis.

- Section 4.1: I find it rather hard to get an overview of the differences between the LI and Aon estimates only based on the scatter plots and correlations. A possibility is to categorize the result (for example low, medium, high losses) and do a more in-depth analysis of the differences within each category. A table summarizing the national losses as low, medium and high losses based on the model's normalized losses and then show statistics of the proportion of events where the models agree on the same category in each country or some other summarizing statistics beyond correlations would help.

- Section 4.1: Figure 6 clearly shows how the LI estimates are smaller than the Aon estimates for high loss events. But it is not clear how much smaller. The LI estimates is proportional to $(v/v\_98)^3$ but experimenting with other exponents $(v/v\_98)^n$ with $n>3$ in the LI equation would tell us how much the cubic assumption in the LI formulation has to be adjusted for the results to be in line with the Aon results for high loss values. This will inform the reader about the level of adjustment needed for the LI formulation to approach the Aon model results.

- Section 5: The conclusion "Compared to Aon's IF Euro WS model, LI ERA5 shows overall lower loss values" cannot be drawn from the analysis. The loss values are not comparable. Aon models' monetary loss and the LI is just a loss index. The max-min

scaling, rescales the values, but the underlying original values are still not comparable.

- Section 5: The conclusion "the Aon model seems to better distinguish between high and moderate impact events" is not justified by the analysis. As the models are not compared to reality we do not know if the Aon model does a "better" job in distinguishing the events, we just know that it separates the loss values between the different events more than the LI estimates.

- Figure 8: Adjust colour scale to better distinguish the different values.

- Section 5: The summarizing list of findings is rather unprecise. Wording such as "comparable behaviour", "slightly shifted", "ranks are comparable" are not very informative.

- Section 5: The authors mention the 72-hour event definition in LI as a possible source of the differences between LI and Aon. Could this be investigated by changing the 72-hour event definition in LI to a 24 hours?

- Section 5+Abstract: I fail to see those conclusions on quality such as "… the loss distribution in LI is not steep enough …" can be justified from the current analysis. What is shown is that it is less steep than the loss distribution of the Aon model. There is no comparison against reality in the paper so we cannot know if it steep enough or not. Recent windstorms like Ciarán/Emir (2023), that is not used for calibration in Aon's model could have been used to shed light on the quality of the loss estimates.

- Section 5: The authors states that the "LI index is missing a detailed damage component, thus it struggles to capture the non-linear response of the buildings at the tail of the gust spectrum for the high impact events." Is not non-linearity what the cubic relation in the LI expression is trying to achieve? The way I see it, is that it is not the lack of non-linearity, but that the non-linearity is less strong than in the Aon model.

- Section 5: A main conclusion for the LI estimates is that "Although it cannot be used to price a storm (due to the missing vulnerability information), it is suitable for estimating the impacts and rank events." It is not clear why it is judged as "suitable". What was the benchmark for the suitability conclusion. What was needed for the estimates to be judged as unsuitable and how do we know it is suitable when it is not compared to reality?

---

## Referee Comment (RC3)

**Insurance loss model vs meteorological loss index – How comparable are their loss estimates for European windstorms?**

By Moemken et al.

The paper compares loss estimates for European windstorms from a simple meteorological Loss Index (LI) with losses from a catastrophe model from model provider Aon Impact Forecasting. The paper aims to shed light on a) the sensitivity of the loss index to different underlying re-analysis data sets and b) the differences between the loss index and the more complex cat model results. The authors conclude that there are substantial differences between the ERA5 and ERA-interim loss index values and that the loss index values have difficulties in distinguishing between extreme and moderate windstorms in terms of loss numbers compared to the insurance loss model.

**General comments**

Given the high importance of natural catastrophes for society, especially in the light of climate change, this topic is clearly within the scope of NHESS. The paper is well-structured and is clearly written. The research questions are clearly posed in the final part of the introduction. The paper has the potential to shed light on the performance of a simple loss index which can be applied to both re-analysis data sets and climate model outputs and the dependency on e.g., spatial resolution. Furthermore, how well such a simple approach can be used to benchmark more refined commercial cat models. This can ultimately help to increase resilience by better understanding of past and future risk of European windstorms.

Overall, the analysis performed appear to be a bit simplistic, though, with mainly correlation and scatter plot analysis between the two meteorological loss indices and the losses from the AON cat model. Most of the analysis do not shed light on the cause of the differences and especially on the quality of the approaches. To answer the research question how comparable and how sensitive the approaches are, more in-depth and refined approaches are recommended.

The comparison between the transparent meteorological based indices and the AON insurance loss models is severely hampered by the fact that the AON model is basically a black box in this analysis. The differences between the approaches likely strongly depend on the vulnerability assumption/ damage functions applied. Here only generic information is given for the AON model in the paper. Therefore, no statements about quality of the approaches can be derived.

In my view the manuscript requires major work to expand the depth of analysis and make the conclusions more stringent.

**Specific comments**

- Abstract: There are a number of qualitative statements such as "comparable storm ranks", "yet it is an effective index", etc, which should be underpinned with quantitative numbers/measures

- Introduction

    1. Given the high interest of climate change, a short statement/references to recent trends of European windstorms should be added

    2. The paper is mainly focusing on the wind-damage/loss relationships which is also one major conclusion from the comparison of the LI and the AON model output. However, loss datasets are not used in the current paper. Therefore I suggest to either expand the scope of the paper and compare the loss estimates to loss datasets or to shorten the introduction in this respect and more focus on wind-damage relationships in the literature

- Section 2.1.

    1. The LI equation (1) is summing up grid squares. One major reason for the differences between ERA5 and ERA-interim results are the different horizonal resolutions as stated by the authors later in the paper - which is kind of superficial. Suggest to exclude this effect in the analysis by appropriate measures.

    2. Ln100: No rationale is given for the threshold chosen. Why 5 storms per season on average? How much % of historic losses of European windstorms are covered by this selection?

    3. Ln109: see above, since the formula (1) is summing up every grid point, differences are to be expected for different horizonal resolutions. This effect should be normalized.

    4. Ln 119: What would be the difference /effect if a 24-hour period is used, similar to the AON approach?

- Section 2.2.

    1. The AON model is only described rather high level. Especially the wind-damage relationship function is key in comparing the results with the LI index and derive meaningful conclusions. Without further details the conclusions will be rather qualitative and vague.

    2. Ln142: Commonly used damage functions assume either a power law or an exponential form. Please discuss why the $v^3$ approach was chosen and discuss the strength and weaknesses of the approach compared to the "commonly used" ones.

    3. Ln148: The hazard component consists of 26 historical events. What is the meteorological data used to define these events? In the table only date and name

are given. How do the wind gust footprints compare between AON and ERA5?

4. Ln150: Was any downscaling performed to derive wind gust at higher horizonal resolution than native resolution of the ECHAM 5 global climate model?

5. Ln164: See above, can more details be given about e.g. one damage curve used in the analysis?

- Section 3.1.

1. The 98$^{th}$ percentile acts quasi as a representation of building codes/standards for the LI approach. Does the AON model also have building code regions (where a similar wind speed would cause a different loss) implemented and if yes, how does the 98$^{th}$ "building code region" pattern compare to these?

- Section 3.2

1. Table1: Correlation numbers for the loss seem to be mis-aligned with Figure 3, e.g. France 0.9059 vs 0.62

2. Ln205: "unlike ERA-interim, ERA5 shows a broader area of high wind guest, especially over the UK and Western continental Europe". Despite, the outlier shows very low LI ERA5 values?

3. Ln215: The obvious reason is the higher spatial resolution of ERA 5: as stated above: to make the analysis more revealing, it is suggested to aim to remove the grid square # dependency by normalization/scaling in the LI formula (1) and look at the remaining differences. Then also absolute values can be used in the analysis of Figure 3.

4. Ln219: ranking of storms: Does this rank analysis really add significant value? At least some rationale for the differences should be given. Consider to instead translate the LI index values in monetary amounts by using one recent storm as reference loss/by normalizing it with e.g. PERILS loss estimate. This would be more tangible and can be also used to compare to AONs estimates (if available) in section 4.

- Section 4

1. Table2: Suggest to add the number of storms for the ERA5 data set to allow for a comparison to the AON model

- Section 4.1.

1. The core analysis of the paper (as also stated in the title) is to reveal the differences between the LI and AON model loss estimates which is mainly addressed in a scatter plot analysis. It is obvious that there are substantial differences between the two approaches. More emphasis should be put on revealing the reasons for the differences seen. Most likely this is due to the different vulnerability curve shape of the $v^3$ approach and the AON model. Since the $v^3$ approach was used in literature quite often in the past, it would be very beneficial to work out the limitations and suggest improvements based on the learnings of the comparison. However, this

would likely mean to have more insights in the AON approach and to cross-check with real world loss numbers.

2. Figure 6 shows clearly the very different behavior of both approaches. Maybe a more in depth analysis for some few selected storms can help to shed more light on the differences, e.g. by looking into the loss contribution to the overall loss by wind speed.

- Section 5

1. "For all of Europe, LI values are higher for ERA5 than for ERA-interim". This is mostly an effect of different horizontal resolution. Suggest to remove and discuss the residual differences as stated above.

2. "Compared to AON's IF model, LI ERA5 shows overall lower loss values". This statement is only true for normalized values but not for (more relevant) monetary values. For this (highly valuable) analysis, LI loss index values need to be translated in monetary loss values and compared to AON's output.

3. Ln296: "the AON model seems to better distinguish between high and moderate impact events": Without benchmark with real world loss numbers this statement appears quite subjective.

4. Ln 299: …, the catastrophe model shows a clear regional dependency of loss values. This regional dependence in less pronounced in LI ERA5. Suggest to discuss the reason for this behavior. Are AON footprints downscaled or have higher resolution?

5. Ln316: As stated above, what is the impact of using a 24h definition also for the LI approach?

6. Ln333: "the LI index is missing a detailed damage component, thus struggles to capture the non-linear response of the buildings at the tail of the gust spectrum for high impact events". The LI $v^3$ approach is obviously non-linear, so the question is what stronger non-linearity would be more suitable. But foremost, from the analysis it is only clear how the wind-loss relationship compares to the unknown AON approach and it is difficult to draw conclusion on the quality of the approaches without comparison to reality.

7. Ln343: "…it is suitable for estimating the impact…" Without more quantitative measure and real world comparisons (and given the large differences to the supposedly more sophisticated AON approach) it is hard to follow this conclusion.

---

## Author Comment (AC1)

We thank Gerard van der Schrier for his comments and suggestions, which helped to improve the manuscript and to remove ambiguities/misunderstandings. Below are point-to-point responses to each comment, including plans how to incorporate them in our manuscript.

**Major concern**

Line 105-110. There is a bit of concern on the parameterization of wind gusts from the reanalysis data. One approach is to use the local near-surface wind speed and its standard deviation in order to estimate the gust (like the Panofsky et al 1977 approach used in the manuscript). This approach makes use of similarity theory, and relates the gust to the friction velocity. The approach performs well in flat terrain, but is sensitive to the parametrization of the local roughness length. The accuracy of the estimated gusts relies heavily on the roughness map that is used, especially when the resolution of the NWP model increases and detailed information about the land-use (and the associated roughness lengths) is required. Errors in the supplied roughness lengths will directly influence the calculated gusts, which is a disadvantage of this approach.

My suggestion is to add a brief analysis where the Panofsky et al (1977) approach is compared to an alternative approach which is specifically suitable for use in a reanalysis product (van den Brink 2019). It links the 1-hour wind speeds at height (which is a standard output of the reanalysis) to 10m wind gusts. This comparison can be done over the entire domain or for specific storms.

To complete this analysis, actual observations of wind gusts should be combined to this assessment. Wind gust values for Europe can be obtained from the European Climate Assessment & Dataset at www.ecad.eu. If you have troubles finding the right data, simply contact ECAD staff.

This additional analysis assess the quality of the parameterization used for the wind gust calculations which is central to this study. The quality of the parameterization is therefore essential and requires a bit more scrutiny that the brief comments that is currently found in the manuscript.

Answer: We believe the reviewer may have misunderstood us here. We did not apply a wind gust parameterization ourselves. Instead, we use the officially published wind gust data from ERA5 and ERA-Interim, which is based on the parameterization approach by Panofsky et al. (1977) and Bechthold & Bidlot (2009). In the present study, we compare for the first time a simple meteorological index to the output of a full insurance windstorm model. Therefore, the assessment of the quality of wind gust parameterization itself is not the aim in our study. However, we would like to note that we have performed such assessments before in dedicated studies, e.g. by comparing different wind gust estimation methods (Born et al., 2012) and by comparing wind gust data from reanalysis to station observations (Seregina et al., 2014). Other authors (e.g. Minola et al., 2020) have also performed such analysis. We will clarify this in the revised manuscript and include some discussion on the effect of wind gust parameterization on the differences between LI and the Aon model output.

**Other aspects**

Section 3.1: fig. 2c shows that ERA5 generally has higher wind gust values than ERA-Interim, but the strongest differences are found over area with complex topography, like the Pyrennees, Alps, Norwegian coast and (perhaps) the Scottish highlands. The higher values over Europe are likely related to a mix of better physics and higher spatial resolution - as the authors correctly state. Could

you explicitly state the spatial resolutions of ERA5 and ERA-Interim in the section where they are introduced? Now the resolution of ERA-Interim is mentioned on line 308 in the very last section of the manuscript. With a coarser resolution, complex topography will be much less well represented and peaks and vallayes will be less pronounced which directly affects the wind gust. Perhaps good to make this explicit in the discussion of the reanalysis.

Answer: The spatial resolution of the different datasets is already stated in the Data and Methods section – in lines 104 and 106, respectively. We agree with the reviewer's comment that the effect of the different spatial resolutions can be important. Thus, we have decided to add some analysis (and corresponding discussion) with using ERA5 re-gridded to the ERA-Interim grid. See also detailed comment further below.

Line 204-206. I am afraid that I fail to see why storm Irina is such an outlier. The Loss Index for ERA-Interim is for this storm much larger than for ERA5, but the storm footprint (fig. S1) does not really show a much larger region where the footprint != 0 over the UK. ERA5 does show higher values (mostly because of the high resolution of ERA5 I guess). So, what do I fail to see in the explanation?

Answer: We thank the reviewer for this comment. The footprint for Irina is overall flatter in ERA5 compared to ERA-Interim. This is particularly the case for the UK, where the mean wind gust over land is 12.1 m/s for ERA5 and 24.6 m/s for ERA-Interim. Therefore, the LI for ERA-Interim is higher due to the cumulative effect (summation of $v/v_{98}$). Additionally, Figure RC1.1 shows an extension of Figure S1 including another panel showing the ERA5 footprint re-gridded to the ERA-Interim resolution. This confirms the overall flatter structure of footprint. We agree this is an important point and we will add more details and Figure RC1.1 in the revised manuscript.

[Figure]

*Figure RC1.1: Wind gust footprint for storm Irina in October 2002 based on ERA5 (a), ERA5 re-gridded to the ERA-Interim grid (b), and ERA-Interim (c). Shown is the largest exceedance (in percent) of the 98th percentile of daily maximum wind gust within 72 hours. The red line and dots denote the cyclone track derived from ERA5 (a, b) and ERA-Interim (c) using the tracking algorithm of Pinto et al. (2005).*

Figure 3 and 4: it would be interesting to add an analysis where ERA5 is first regridded to the ERA-Interim resolution, and then the LI diagrams are made. This analysis gives a clue if it is the improved physics in ERA5 that makes the difference or that the increase in spatial resolution makes the difference. This would be nice to add to the Supplementary material. This analysis could then provide the basis for Section 5, bullet 1: I have not seen evidence that it is the resolution that makes ERA5 better than ERA-Interim (although this is likely).

Answer: We thank the reviewer for this comment. Figure RC1.2 shows the LI for ERA5 re-gridded to the ERA-Interim resolution for different regions, compared to the original Figure S2. After re-gridding, LI ERA5 and LI ERA-Interim are in the same order of magnitude, while the overall behavior/order of storms does not change (as can be seen by the small changes in $R^2$). This confirms that the different resolution of the datasets is not decisive – and that differences may most likely result from differences in the wind gust distribution (see also Figure RC1.3). We will add the right part of Figure RC1.2 to the Supplementary and enhance the discussion in the revised manuscript.

[Figure]

**Figure RC1.2:** *Comparison of loss values (in thousands) based on LI ERA5 (x-axis) and LI ERA-Interim (y-axis) for the common 20 most extremes storms in the period 1979-2019. LI ERA5 is calculated from the original ERA5 gust data (left) and from the ERA5 gust data re-gridded to the ERA-Interim resolution (right).*

Section 5: bullet 1: the distribution of wind gusts may be shifted right in ERA5, but the footprint uses the 98th percentile - which is also shifted right. So this argument does not make sense.

Answer: The reviewer is correct here – a shift in the wind gust distribution also implies a shift in the 98th percentile and therefore cannot explain the higher LI values for ERA5 alone. What we meant is that besides the overall shift in the distribution, the tail of the distribution is longer for ERA5, which then leads to higher LI values. To illustrate this, Figure RC1.3 shows both the 98th and the 99.9th percentile for ERA5 and ERA-Interim as well as the difference. The figure clearly shows larger differences for higher (99.9th) percentiles, thereby confirming the longer tail of the distribution in ERA5. We apologize for the misunderstanding. We will replace Figure 2 by Figure RC1.3 and clarify this point in the revised paper.

[Figure]

***Figure RC1.3:*** *98th percentile (upper row) and 99.9th percentile (lower row) of daily maximum wind gust for the winter half year (October – March, ONDJFM) for the period 1979-2019.*

Figure S1: In the caption of the figure you write "Shown is the percentage of the maximum wind gust in 72 hours that exceed the 98th percentile of daily maximum wind gust." If you aim to show the outcome of equation 2, the this should be something like "Shown is the strength of the maximum wind gust in 72 hours as deviation from the 98th percentile and normalized with the 98th percentile."

Answer: We apologize for the misleading figure caption. We will change it to "Shown is the largest exceedance (in percent) of the 98th percentile of daily maximum wind gust within 72 hours" in the revised manuscript.

References

Bechtold, P. and Bidlot, J. R.: Parametrization of convective gusts, ECMWF Newsletter, 199, 15-18, https://doi.org/10.21957/kfr42kfp8c, 2009

Born, K., Ludwig, P., and Pinto, J. G.: Wind gust estimation for Mid-European winter storms: towards a probabilistic view, Tellus A, 64, https://doi.org/10.3402/tellusa.v64i0.17471, 2012

Minola, L., Zhang, F., Azorin-Molina, C. et al. : Near-surface mean and gust wind speeds in ERA5 across Sweden: towards an improved gust parametrization, Clim. Dyn., 55, 887-907, https://doi.org/10.1007/s00382-020-05302-6, 2020

Panofsky, H. A., Tennekes, H., Lenschow, D. H., and Wyngaard, J. C.: The characteristics of turbulent velocity components in the surface layer under convective conditions, Bound.-Lay. Meteorol., 11, 355-361, https://doi.org/10.1007/BF02186086, 1977

Seregina, L. S., Haas, R., Born, K., and Pinto, J. G.: Development of a wind gust model to estimate gust speeds and their return periods. Tellus A, 66, https://doi.org/10.3402/tellusa.v66.22905, 2014

---

## Author Comment (AC2)

We thank the reviewer for his/her detailed and constructive comments, which helped to improve the manuscript and clarify key points. Below are point-to-point responses to each comment, including plans how to incorporate them in our manuscript.

**General comment**

It is clearly within the scope of NHESSS and is written in a clear and well-structured way. The research questions are clearly outlined in the final part of the introduction. The paper has the potential to shed light on the differences between a rather simple, but well documented open access approach and a more refined proprietary commercial product. This is very welcome contribution that could inform the community on the differences between approaches pursued by the academic community and private sector.

I find the analysis to be somewhat superficial and with few exceptions it consists of correlation analysis and scatter plots. The correlation analysis is rather hard to interpret and to properly answer the research question "How comparable are windstorm loss estimates from this meteorological index and an insurance loss model?" a much more multi-faceted approach is needed.

Answer: We agree with the reviewer that a more detailed analysis would be important to strengthen the scientific values of the manuscript and in particular to understand how comparable LI and the Aon IF model output are. We will include a detailed case study for recent storm Sabine (February 2020) in the updated manuscript, in which we compare LI and the Aon IF model output, and add aggregated market losses from the PERILS data as a benchmark. Figure RC2.1 shows both the normalized losses and the storm ranking at country level for storm Sabine for the three datasets. In the revised manuscript, we will replace Figure 5 with Figure RC2.1. Additionally, we will extend Figure 8 (Spearman rank correlation) by adding PERILS to the comparison (see Figure RC2.2).

[Figure]

*Figure RC2.1: Normalized losses (upper row) and storm ranking (lower row) at country level for storm Sabine in February 2020. Losses are derived from LI ERA5 (left), Aon's IF Euro WS model (middle), and PERILS (right). The black line and dots in the left column denote the cyclone track derived from ERA5 using the tracking algorithm of Pinto et al. (2005). Losses are only shown for the 11 countries covered by Aon. The ranking is based on common storms.*

[Figure]

*Figure RC2.2:* *Spearman's rank correlation coefficient at country level for LI ERA5 vs Aon's IF Euro WS model (left), LI ERA5 vs PERILS (middle), and Aon's IF Euro WS model vs PERILS (right). The ranking is based on common storms per country.*

Aon's Impact Forecasting model is a commercial windstorm model and is treated as a black box. From a scientific point of view this seriously hampers both the depth of the analysis and the information value of the conclusions. Questions like why the two approaches differ can only be answered with statements concerning the shape of the final loss distribution and there is no comparison against reality which prevents any statements on quality.

Answer: We agree with the reviewer in this point. We will provide a more detailed description (roughly 10 pages) of the Aon IF windstorm model, including a description of the hazard, vulnerability and exposure components, which will be included in the Supplementary. Based on this detailed description, we will also enhance the discussion on the methodological differences between the two approaches in the revised manuscript.

I believe the manuscript requires major work before it can be accepted and recommend a resubmission or a major revision.

**Specific comments**

Abstract: Research question two (comparison between the models) which from the title of the paper is the most important only have three sentences in the abstract.

Answer: We thank the reviewer for this comment. We will adapt the abstract so that it better matches the title of our paper.

Abstract: It would be beneficial if the qualitative statement such as "shows comparable storm ranks", "yet it is an effective index", "higher LI values for ERA5 than for ERA-Interim" are justified by some numbers.

Answer: We will justify some of the statements by including actual numbers.

Introduction: The introduction starts out with a rather weak statement "In Central and Western Europe, windstorms are among the major natural hazards". I am sure a more precise description of the importance can be found. For example in https://wmo.int/publication-series/atlas-of-mortality-and-economic-losses-from-weather-climate-and-water-related-hazards-1970-2021

Answer: We started the introduction with a rather general statement to address a broad audience, including a more precise description in the following sentences. Therefore, we would like to keep the statement as it is.

Introduction: There is a rather lengthy paragraph on loss datasets, but very little on different wind-damage functions. Given that the paper is on estimation of loss and particularly the steepness of the loss with increasing winds, I suggest that more emphasis is put on the describing the literature on wind-damage relations and less on the damage datasets (which was reviewed in the authors previous paper).

Answer: We appreciate this comment and the suggestion to focus more on wind-damage relations. We will shorten the paragraph on damage datasets and expand the part on damage functions.

Section 2:1: The LI equation is summing up grid squares without any grid square area scaling. A storm centered at 45N will have grid squares that are around 40% larger than one centered at 60N. Thus, equally sized storms will have rather different number of grid squares, making the LI values not directly comparable. This should be mentioned in the text as a possible weakness of the LI when used over regions spanning a large area in the north-south direction.

Answer: We agree with the reviewer that LI can depend on the size of the grid squares. However, we believe this effect to be negligible for our study, as we mainly focus on individual countries and Core Europe (which has no large expansion in north-south direction). Nevertheless, we will add a short note on this effect in the revised manuscript.

Section 2:1: Only events with LI values above a certain threshold are kept. The authors note that they are only interested in extreme storm events and they use a threshold "which corresponds to the selection of an average of five events per season". Does this mean that if you have 41 years of data there will be 41*5=205 events (assuming that there is only one season in this study, ONDJFM)?

Answer: Yes, the reviewer is correct in this point. We will clarify it in the updated manuscript.

Section 2.1: The post-processing of wind gusts based hourly data is difficult and needs a more detailed description. I wonder if this step is really necessary? Post-processed wind gusts based on hourly data tend to be strongly correlated to the hourly winds and have similar distribution shape. If this is the case the $(v/v\_98)3$ ratio in the Loss Index (LI) should be almost the same if the daily max hourly wind is used instead of the wind gust data. If this is the case, there is no need to introduce a questionable wind gust parameterization, the $(v/v\_98)3$ ratio could be calculated directly from the daily max hourly wind speed.

Answer: We believe the reviewer may have misunderstood us here. We did not apply a wind gust parameterization ourselves. Instead, we use the officially published wind gust data from ERA5 and ERA-Interim, which is based on the parameterization approach by Panofsky et al. (1977) and Bechtold & Bidlot (2009). We will clarify this in the revised manuscript.

Section 2.1: Make a separate section for the input data/information. (ERA, population density data, storm names etc.). It should not be described in the Meteorological loss index section.

Answer: We thank the reviewer for this comment. We will improve the structure of section 2 by clearly separating the data description from the methods.

Section 2.2: The first paragraph is not about the Aon model or the Perils data. Consider removing it.

Answer: The paragraph is intended as a general introduction to loss models. We would therefore like to keep it as it is.

Section 2.2: The Aon model is not described in much detail. Is there no more information available on the model? The paper would benefit strongly from a more detailed description of the Aon model.

Answer: Please refer to our reply on the general comments above.

Section 2.2: Consider making a separate data section where the PERILS data can be described.

Answer: We will improve the structure of section 2 by clearly separating the data description from the methods.

Section 3.1: Section 3 starts with focusing is on the comparison between ERA5 and ERA-Interim 98th percentile wind gust. The 98th percentile is the loss-no loss threshold which is only marginally interesting. As the loss is increasing with $(v/v\_98)3$ a more revealing analysis would be the difference between for example $(v\_99.7/v\_98)3$ (99.7 percentile is the once-a-year value) or $(v\_max/v\_98)3$. I think this would better describe how the differences between ERA5 and ERA-Interim would affect the loss calculations.

Answer: We thank the reviewer for this comment. Figure RC2.3 shows an extension of Figure 2, additionally including the 99.9th percentile for both datasets as well as the difference. The differences between ERA5 and ERA-Interim are larger for higher percentiles – suggesting a longer tail of the wind gust distribution for ERA5, which could result in overall higher LI values. We will replace Figure 2 by Figure RC2.3 and enhance the discussion in the revised manuscript.

[Figure]

*Figure RC2.3: 98th percentile (upper row) and 99.9th percentile (lower row) of daily maximum wind gust for the winter half year (October – March, ONDJFM) for the period 1979-2019.*

Section 3.2: It is rather unfortunate that the losses is max-min normalized when ERA5 and ERA-Interim are compared. Is it not more revealing to do a grid square area scaling to get rid of the grid

square dependence in the LI method and then look at the remaining differences in the Loss Index distributions instead of showing the rescaled versions?

Answer: The original (not normalized) loss values are shown in Supplementary Figure S2. We decided to keep the normalized values in the main paper in order to be consistent with Figure 6 in section 4.1.

We tested the scaling effect of the different spatial resolutions by re-gridding ERA5 to the ERA-Interim grid before calculating LI. Figure RC2.4 shows the LI for ERA5 re-gridded to the ERA-Interim resolution for different regions, compared to the original Figure S2. After re-gridding, LI ERA5 and LI ERA-Interim are in the same order of magnitude, while the overall behavior/order of storms does not change (as can be seen by the small changes in $R^2$). This confirms that the different resolution of the datasets is not decisive – and that differences may most likely result from differences in the wind gust distribution (see also previous comment). We will add the right part of Figure RC2.4 to the Supplementary and enhance the discussion in the revised manuscript.

[Figure]

***Figure RC2.4:*** *Comparison of loss values (in thousands) based on LI ERA5 (x-axis) and LI ERA-Interim (y-axis) for the common 20 most extremes storms in the period 1979-2019. LI ERA5 is calculated from the original ERA5 gust data (left) and from the ERA5 gust data re-gridded to the ERA-Interim resolution (right).*

Section 3.2: The storm losses and storm ranking comparison is done using Pearson and rank correlations. For the Pearson correlation to be informative the data need to be normally distributed. It is pretty clear that the loss data is heavily skewed and the Pearson correlation becomes pretty meaningless or at least very hard to interpret. The R2 will not be the variance the two datasets have in common. Either the data must be transformed to become normally distributed (Box-Cox or similar) or other measures of similarity should be applied. It also seems that the authors have used the paired Wilcoxon Signed-Rank Test on the Pearson correlations. But as the Pearson correlation assumes normal distributed data, the t-test would be the appropriate test.

Answer: We believe the reviewer may have misunderstood us here. We are not using Pearson correlation, but only the coefficient of determination $R^2$ that directly relates to the linear regression lines as shown in Figures 3, 4, 6 & 7. The larger $R^2$ is, the greater the linear relation between the two datasets. However, in order to avoid misunderstandings, we will merge Tables 1 and 2, and will additionally replace the $R^2$ with the one calculated based on Spearman's rank correlation coefficient (see Table RC2.1). Please also refer to one of the following comments.

*Table RC2.1: $R^2$ of Spearman's rank correlation coefficient between LI ERA5 and LI ERA-Interim (2nd column), LI ERA5 and Aon's IF Euro WS model (3rd column), LI ERA5 and PERILS (4th column), and Aon's IF Euro WS model and PERILS (last column). The number of common storms per country is given in brackets.*

|  | LI ERA5 vs LI ERA-Interim | LI ERA5 vs Aon's IF Euro WS | LI ERA5 vs PERILS | Aon's IF Euro WS vs PERILS |
|---|---|---|---|---|
| **Core Europe** | 0.65 [20] | 0.52 [23] | 0.26 [17] | 0.57 [19] |
| **Austria** | 0.43 [20] | 0.75 [15] | 1.0 [4] | 1.0 [4] |
| **Belgium** | 0.62 [20] | 0.22 [21] | 0.09 [11] | 0.66 [11] |
| **Denmark** | 0.25 [20] | 0.41 [15] | 0.49 [5] | 0.14 [6] |
| **France** | 0.79 [20] | 0.6 [17] | 0.56 [10] | 0.54 [11] |
| **Germany** | 0.5 [20] | 0.57 [23] | 0.33 [15] | 0.47 [15] |
| **Ireland** | 0.37 [20] | 0.2 [19] | 0.49 [5] | 0.64 [5] |
| **Luxembourg** | 0.64 [20] | 0.26 [15] | 0.07 [6] | 0.43 [6] |
| **Netherlands** | 0.2 [20] | 0.64 [21] | 0.68 [11] | 0.7 [11] |
| **Norway** | 0.29 [20] | 0.4 [9] | 0.25 [3] | 1.0 [3] |
| **Sweden** | 0.51 [20] | 0.23 [13] | 1.0 [4] | 0.16 [4] |
| **United Kingdom** | 0.49 [20] | 0.36 [20] | 0.44 [13] | 0.7 [13] |

Section 3.2: The authors note that they are "ranking for the 20 common most extreme storms", but it is not necessarily the most extreme storms, but the one with largest loss values.

Answer: As the whole study deals with windstorm losses, we use most extreme in the sense of loss. This does not necessarily mean extreme in terms of wind gust, core pressure minimum or other. We will state this clearly in the updated text.

Section 3.2: How is the "20 common most extreme storms" found and how will this affect the ranking analysis? If a common storm is no. 18 in one dataset and no. 24 in another, but picked as one of the 20 common most extreme storms, will they be reranked (between 1 and 20) and the ranking analysis based on the reranked values? If they are based on reranked values, the results may not be representative for the original datasets.

Answer: In general, the storm ranking depends on the total number of storms per dataset (Moemken et al., 2024). This is particularly relevant for the later comparison between LI and the Aon IF model. Therefore, we decided to first select the 20 common storms and rank only these events. However, in order not to lose the information from the original datasets, we will add the original rank of the events in Tables S2 and S3 in the revised manuscript.

Section 3.2: The authors state that the LI values based on ERA5 are approximately 10 times larger than for ERA-Interim due to smaller grid squares. To highlight possible other reasons, the LI values could be scaled with grid square area, to investigate if any systematic differences that was not due to the obvious grid square size dependence.

Answer: Please refer to one of the previous comments and Figure RC2.4.

Section 3.2: Storm loss rankings are based on shared variance of the ranked scores (rank correlation squared). It is not mentioned in the text, but I guess this is based on the Spearman rank correlation?

Answer: We are not completely sure that we understand this comment. Is the reviewer asking which correlation we use for the values in Table 1? In the current version, the values in Table 1 correspond to the coefficient of determination $R^2$ from the linear regression in Figures 3 and 4. Based on the other comments and in order to avoid misunderstandings, we will change this in the revised manuscript and use $R^2$ values from the Spearman's rank correlation (see Table RC2.1).

Section 3.2: One might wonder if the correlation analysis really is the way is to go for showing differences in the loss estimates? I have a hard time understanding what a R2 of 0.5 really means. According to the paired Wilcoxon Signed-Rank Test correlations squared down to 0.27 and rank correlations squared down to 0.18 does not indicate significant differences between LI ERA5 and LI ERA-Interim. Given the low number of events and possibly a few events that are very different in the two sets, there might be other methods than correlation analysis that are more useful and easier to understand. Maybe categorizing the losses, Kendal tau, Rank Biased Overlap, Goodman and Kruskal's gamma, precision etc. or other methods can be considered.

Answer: Our study is the first to compare a full insurance windstorm model (which is not available publicly) to a simplified meteorological loss index. Therefore, we decided to focus on a straightforward comparison of the two methods. With this aim, we decided to use the Spearman R and $R^2$ coefficients, which allows us to focus on simple but robust conclusions and to avoid a mix of different statistical methods.

Section 4.1: The same comments on the use of Pearson correlation etc. mentioned in section 3.2 applies on this analysis.

Answer: Please refer to our reply on the comment regarding section 3.2

Section 4.1: I find it rather hard to get an overview of the differences between the LI and Aon estimates only based on the scatter plots and correlations. A possibility is to categorize the result (for example low, medium, high losses) and do a more in-depth analysis of the differences within each category. A table summarizing the national losses as low, medium and high losses based on the model's normalized losses and then show statistics of the proportion of events where the models

agree on the same category in each country or some other summarizing statistics beyond correlations would help.

Answer: We agree with the reviewer that there may be merit in the idea of investigating the different behavior of high and low loss events. However, as the sample size is already quite small (see Table RC2.1), we might have issues with splitting the data into even smaller samples. Therefore, we decided to analyze the sample as a whole.

Section 4.1: Figure 6 clearly shows how the LI estimates are smaller than the Aon estimates for high loss events. But it is not clear how much smaller. The LI estimates is proportional to $(v/v\_98)3$ but experimenting with other exponents $(v/v\_98)n$ with $n>3$ in the LI equation would tell us how much the cubic assumption in the LI formulation has to be adjusted for the results to be in line with the Aon results for high loss values. This will inform the reader about the level of adjustment needed for the LI formulation to approach the Aon model results.

Answer: The LI method used in our study is well established. Based on the original approach developed for station data from Klawa & Ulbrich (2003), it was further developed by Pinto et al. (2012), and several formulations (also with other exponents) were tested. The same was done in other studies such as Prahl et al. (2015). All these studies agree that the performance of the different indices depends on the underlying event set. For some storm events, formulations with higher exponents seem to better suit to realistically estimate windstorm losses, while for other events, the cubic relationship provides results that are more realistic. In this sense, and based in our experience, no formulation clearly outperforms the others. Since our study is the first to compare a full insurance windstorm model (which is not available publicly) to a simplified meteorological loss index, we focus on a straightforward comparison of the two methods (as mentioned before). Therefore, in our opinion, the objective should not be to experiment with the LI formulation. Nevertheless, we will provide a more detailed discussion of the impact of the LI setup on the results in the revised manuscript.

Section 5: The conclusion "Compared to Aon's IF Euro WS model, LI ERA5 shows overall lower loss values" cannot be drawn from the analysis. The loss values are not comparable. Aon models' monetary loss and the LI is just a loss index. The max-min scaling, rescales the values, but the underlying original values are still not comparable.

Answer: We agree that the original loss values of LI and the Aon IF model are not comparable. Based on the normalized losses, however, we do think we are able to draw conclusions such as the one referred to in the comment. Nevertheless, we will rephrase the conclusion to avoid confusion.

Section 5: The conclusion "the Aon model seems to better distinguish between high and moderate impact events" is not justified by the analysis. As the models are not compared to reality we do not know if the Aon model does a "better" job in distinguishing the events, we just know that it separates the loss values between the different events more than the LI estimates.

Answer: We agree with the reviewer in this point. The Aon IF model is calibrated against the PERILS dataset, thereby it can be assumed as a representation of a "market-reality" for the purpose of this paper. Nevertheless, we decided to use none of the datasets as ground truth (Moemken et al., 2024). We will carefully go through section 5 and rephrase sentences where necessary.

Figure 8: Adjust colour scale to better distinguish the different values.

Answer: Thanks for pointing this out. We will adjust the color scale in the revised manuscript.

Section 5: The summarizing list of findings is rather unprecise. Wording such as "comparable behaviour", "slightly shifted", "ranks are comparable" are not very informative.

Answer: We will make the summary more precise in the revised paper.

Section 5: The authors mention the 72-hour event definition in LI as a possible source of the differences between LI and Aon. Could this be investigated by changing the 72-hour event definition in LI to a 24 hours?

Answer: We appreciate this suggestion. We did a sensitivity analysis for different time windows, e.g. calculating LI for 24-hour windows. Figure RC2.5 shows the results for the normalized losses and figure RC2.6 for the storm ranking, respectively. Overall, we find no systematic reduction in the differences between LI and the Aon IF model output when using 24 hours instead of 72 hours. For some storms and/or countries, the correlations increase with a shorter event definition (see e.g. Germany), while for others they decrease (see e.g. Core Europe). In addition, the number of common storms decreases when using 24-hour windows for the LI calculation (not shown). Therefore, we decided to keep the focus of our study on the 72-hour event definition. This has several advantages: First, we are able to capture the entire storm footprint; second, this is in line with the standard practice in insurance industry (the so-called 72-hour-clause); third, the correlations between LI and the Aon IF model are higher, especially for Core Europe. Nevertheless, we will include both figures in the Supplementary and expand the discussion in the revised manuscript.

[Figure]

*Figure RC2.5: Comparison of normalized loss values between Aon's IF Euro WS model (x-axis) and LI ERA5 (y-axis). Depicted are the common most extreme storms for the period 1990-2020 for (a) Core Europe, (b) the United Kingdom, (c) Germany, and (d) France. A logarithmic scale is used for the axes. The red dashed line denotes the logarithmic regression. The correlation between the datasets is given in the upper left corner ($R^2$ value). Outlier storms based on the IQR method are marked in red. LI ERA5 is calculated for 72-hour windows (left) and 24-hour windows (right).*

[Figure]

**LI ERA5 (72 hours)**    **LI ERA5 (24 hours)**

*Figure RC2.6: Same as Figure RC2.5, but for the comparison of storm ranks. The values in brackets indicate the rank (first value Aon's model, second value LI ERA5).*

Section 5+Abstract: I fail to see those conclusions on quality such as "… the loss distribution in LI is not steep enough …" can be justified from the current analysis. What is shown is that it is less steep than the loss distribution of the Aon model. There is no comparison against reality in the paper so we cannot know if it steep enough or not. Recent windstorms like Ciarán/Emir (2023), that is not used for calibration in Aon's model could have been used to shed light on the quality of the loss estimates.

Answer: We agree with the reviewer that it can be complex to explain some of our conclusions from the analysis. The Aon IF model is calibrated against real loss data, using the PERILS data as the primary benchmark. Therefore, all storms considered in our analysis are calibrated/validated against a "market-reality". We will clarify this in the revised manuscript. Additionally, we will include the PERILS data in some of the analysis (see above).

Section 5: The authors states that the "LI index is missing a detailed damage component, thus it struggles to capture the non-linear response of the buildings at the tail of the gust spectrum for the high impact events." Is not non-linearity what the cubic relation in the LI expression is trying to achieve? The way I see it, is that it is not the lack of non-linearity, but that the non-linearity is less strong than in the Aon model.

Answer: We agree with the reviewer and will reword the statement in the revised manuscript (see also previous comment).

Section 5: A main conclusion for the LI estimates is that "Although it cannot be used to price a storm (due to the missing vulnerability information), it is suitable for estimating the impacts and rank events." It is not clear why it is judged as "suitable". What was the benchmark for the suitability conclusion. What was needed for the estimates to be judged as unsuitable and how do we know it is suitable when it is not compared to reality?

Answer: We agree with the reviewer in this point. We will reword the statement in the revised manuscript – taking into account the fact that we propose to use the Aon IF model output as a representation of a "market-reality" for the purpose of this study (see also the second last comment).

References

Bechtold, P. and Bidlot, J. R.: Parametrization of convective gusts, ECMWF Newsletter, 199, 15-18, https://doi.org/10.21957/kfr42kfp8c, 2009

Klawa, M. and Ulbrich, U.: A model for the estimation of storm losses and the identification of severe winter storms in Germany, Nat. Hazards Earth Syst. Sci., 3, 725-732, https://doi.org/10.5194/nhess-3-725-2003, 2003

Moemken, J., Messori, G., and Pinto, J. G.: Windstorm losses in Europe – What to gain from damage datasets, Weather and Climate Extremes 44, 100661, https://doi.org/10.1016/j.wace.2024.100661, 2024

Panofsky, H. A., Tennekes, H., Lenschow, D. H., and Wyngaard, J. C.: The characteristics of turbulent velocity components in the surface layer under convective conditions, Bound.-Lay. Meteorol., 11, 355-361, https://doi.org/10.1007/BF02186086, 1977

Pinto, J. G., Karremann, M., Born, K., Della-Marta, P., and Klawa, M.: Loss potentials associated with European windstorms under future climate conditions, Climate Res., 54, 1-20, https://doi.org/10.3354/cr01111, 2012

Prahl, B. F., Rybski, D., Burghoff, O., and Kropp, J. P.: Comparison of storm damage functions and their performance, Nat. Hazards Earth Syst. Sci., 15, 769-788, https://doi.org/10.5194/nhess-15-769-2015, 2015

---

## Author Comment (AC3)

We thank the reviewer for his/her constructive comments, which helped to improve the manuscript and clarify important points. Below are point-to-point responses to each comment, including plans how to incorporate them in our manuscript.

**General comments**

Given the high importance of natural catastrophes for society, especially in the light of climate change, this topic is clearly within the scope of NHESS. The paper is well-structured and is clearly written. The research questions are clearly posed in the final part of the introduction. The paper has the potential to shed light on the performance of a simple loss index which can be applied to both re-analysis data sets and climate model outputs and the dependency on e.g., spatial resolution. Furthermore, how well such a simple approach can be used to benchmark more refined commercial cat models. This can ultimately help to increase resilience by better understanding of past and future risk of European windstorms.

Overall, the analysis performed appear to be a bit simplistic, though, with mainly correlation and scatter plot analysis between the two meteorological loss indices and the losses from the AON cat model. Most of the analysis do not shed light on the cause of the differences and especially on the quality of the approaches. To answer the research question how comparable and how sensitive the approaches are, more in-depth and refined approaches are recommended.

Answer: We agree with the reviewer that a more detailed analysis would be important to strengthen the scientific values of the manuscript and in particular to understand how comparable LI and the Aon IF model output are. We will include a detailed case study for recent storm Sabine (February 2020) in the updated manuscript, in which we compare LI and the Aon IF model output, and add aggregated market losses from the PERILS data as a reference. Figure RC3.1 shows both the normalized losses and the storm ranking at country level for storm Sabine for the three datasets. In the revised manuscript, we will replace Figure 5 with Figure RC3.1. Additionally, we will extend Figure 8 (Spearman rank correlation) by adding PERILS to the comparison (see Figure RC3.2).

[Figure]

***Figure RC3.1:*** *Normalized losses (upper row) and storm ranking (lower row) at country level for storm Sabine in February 2020. Losses are derived from LI ERA5 (left), Aon's IF Euro WS model (middle), and PERILS (right). The*

*black line and dots in the left column denote the cyclone track derived from ERA5 using the tracking algorithm of Pinto et al. (2005). Losses are only shown for the 11 countries covered by Aon. The ranking is based on common storms.*

[Figure]

**Figure RC3.2:** *Spearman's rank correlation coefficient at country level for LI ERA 5 vs Aon's IF Euro WS model (left), LI ERA5 vs PERILS (middle), and Aon's IF Euro WS model vs PERILS (right). The ranking is based on common storms per country.*

The comparison between the transparent meteorological based indices and the AON insurance loss models is severely hampered by the fact that the AON model is basically a black box in this analysis. The differences between the approaches likely strongly depend on the vulnerability assumption/ damage functions applied. Here only generic information is given for the AON model in the paper. Therefore, no statements about quality of the approaches can be derived.

Answer: We agree with the reviewer in this point. We will provide a more detailed description (roughly 10 pages) of the Aon IF windstorm model, including a description of the hazard, vulnerability and exposure components, which will be included in the Supplementary. Based on this detailed description, we will also enhance the discussion on the methodological differences between the two approaches in the revised manuscript.

In my view the manuscript requires major work to expand the depth of analysis and make the conclusions more stringent.

**Specific comments**

Abstract: There are a number of qualitative statements such as "comparable storm ranks", "yet it is an effective index", etc, which should be underpinned with quantitative numbers/measures

Answer: We will underpin some of the statements by including actual numbers.

Introduction

1.  Given the high interest of climate change, a short statement/references to recent trends of European windstorms should be added
    Answer: The manuscript is not about trends or decadal variability of windstorm activity in Europe. However, we will add a statement that decadal and longer variability may be present in the datasets for the historical period (e.g. Feser et al., 2015).

2.  The paper is mainly focusing on the wind-damage/loss relationships which is also one major conclusion from the comparison of the LI and the AON model output. However, loss datasets are not used in the current paper. Therefore I suggest to either expand the scope of the

paper and compare the loss estimates to loss datasets or to shorten the introduction in this respect and more focus on wind-damage relationships in the literature

Answer: We thank the reviewer for pointing this out. We will shorten the paragraph on damage datasets and expand the part on damage functions.

Section 2.1.

1. The LI equation (1) is summing up grid squares. One major reason for the differences between ERA5 and ERA-interim results are the different horizonal resolutions as stated by the authors later in the paper - which is kind of superficial. Suggest to exclude this effect in the analysis by appropriate measures.

   Answer: We tested the scaling effect of the different spatial resolutions by re-gridding ERA5 to the ERA-Interim grid before calculating LI. Figure RC3.3 shows the LI for ERA5 re-gridded to the ERA-Interim resolution for different regions, compared to the original Figure S2. After re-gridding, LI ERA5 and LI ERA-Interim are in the same order of magnitude, while the overall behavior/order of storms does not change (as can be seen by the small changes in $R^2$). This confirms that the different resolution of the datasets is not decisive – and that differences may most likely result from differences in the wind gust distribution (see also previous comment). We will add the right part of RC3.3 to the Supplementary and enhance the discussion in the revised manuscript.

[Figure]

*Figure RC3.3: Comparison of loss values (in thousands) based on LI ERA5 (x-axis) and LI ERA-Interim (y-axis) for the common 20 most extremes storms in the period 1979-2019. LI ERA5 is calculated from the original ERA5 gust data (left) and from the ERA5 gust data re-gridded to the ERA-Interim resolution (right).*

2. Ln100: No rationale is given for the threshold chosen. Why 5 storms per season on average? How much % of historic losses of European windstorms are covered by this selection?

   Answer: From our experience, we typically have a maximum of 3-5 important windstorms per season (in terms of insurance losses and impact on the European market). Given that we

are focusing on the top 20-25 storms for a period of 41 years most of the time, we do not think this pre-selection is a large constraint.

3. Ln109: see above, since the formula (1) is summing up every grid point, differences are to be expected for different horizonal resolutions. This effect should be normalized.
Answer: Please refer to our reply on one of the above comments.

4. Ln 119: What would be the difference /effect if a 24-hour period is used, similar to the AON approach?
Answer: We appreciate this suggestion. We did a sensitivity analysis for different time windows, e.g. calculating LI for 24-hour windows. Figure RC3.4 shows the results for the normalized losses and figure RC3.5 for the storm ranking, respectively. Overall, we find no systematic reduction in the differences between LI and the Aon IF model output when using 24 hours instead of 72 hours. For some storms and/or countries, the correlations increase with a shorter event definition (see e.g. Germany), while for others they decrease (see e.g. Core Europe). In addition, the number of common storms decreases when using 24-hour windows for the LI calculation (not shown). Therefore, we decided to keep the focus of our study on the 72-hour event definition. This has several advantages: First, we are able to capture the entire storm footprint; second, this is in line with the standard practice in insurance industry (the so-called 72-hour-clause); third, the correlations between LI and the Aon IF model are higher, especially for Core Europe. Nevertheless, we will include both figures in the Supplementary and expand the discussion in the revised manuscript.

[Figure]

*Figure RC3.4:* *Comparison of normalized loss values between Aon's IF Euro WS model (x-axis) and LI ERA5 (y-axis). Depicted are the common most extreme storms for the period 1990-2020 for (a) Core Europe, (b) the United Kingdom, (c) Germany, and (d) France. A logarithmic scale is used for the axes. The red dashed line denotes the logarithmic regression. The correlation between the datasets is given in the upper left corner (R2 value). Outlier storms based on the IQR method are marked in red. LI ERA5 is calculated for 72-hour windows (left) and 24-hour windows (right).*

[Figure]

***Figure RC3.5:*** *Same as Figure RC3.4, but for the comparison of storm ranks. The values in brackets indicate the rank (first value Aon's model, second value LI ERA5).*

Section 2.2.

1. The AON model is only described rather high level. Especially the wind-damage relationship function is key in comparing the results with the LI index and derive meaningful conclusions. Without further details the conclusions will be rather qualitative and vague.
   Answer: Please refer to our reply on the general comments above.

2. Ln142: Commonly used damage functions assume either a power law or an exponential form. Please discuss why the v3 approach was chosen and discuss the strength and weaknesses of the approach compared to the "commonly used" ones.
   Answer: We thank the reviewer for this comment. We will provide a more detailed discussion of the impact of the LI setup on the results in the revised manuscript.

3. Ln148: The hazard component consists of 26 historical events. What is the meteorological data used to define these events? In the table only date and name are given. How do the wind gust footprints compare between AON and ERA5?
   Answer: For historic modelling, footprints are built directly from weather station data. Inverse distance weighting is used to interpolate station values to the model grid at 7 km resolution. The gridded footprints are then implemented in the Aon IF model for loss calculation. Depending on the date and the geography of the specific event, a variety of data sources has been used to build the footprints – primary WMO station data, but also data from national meteorological services such as the British Meteorological Office (UKMO), the Danish Meteorological Institute (DMI), the Dutch Meteorological Institute (KNMI), the Irish Meteorological Service (Met Éireann), the Swedish Meteorological and Hydrological Institute (SMHI) the Finish Meteorological Institute (FMI), and the Norwegian Meteorological Institute (NMI). We will provide a detailed description in the revised Supplementary.

4. Ln150: Was any downscaling performed to derive wind gust at higher horizontal resolution than native resolution of the ECHAM 5 global climate model?

Answer: The stochastic model consists of 12,044 simulated storms. These have been extracted from the ECHAM5 Global Circulation Model. More information about the generation of this dataset is given in Karremann et al. (2014). The resolution of the pre-downscaled stochastic storms is 1.875° x 1.875°, approximately 200 x 200 km in the mid-latitudes. The stochastic storms are calibrated against a set of 124 historic storms taken from NCEP reanalysis data. A combination of dynamical downscaling with COSMO-CLM and statistical downscaling is used to produce the final high-resolution stochastic event set implemented in the Aon IF Euro WS model. A detailed description will be made available in the Supplementary of the revised manuscript.

5. Ln164: See above, can more details be given about e.g. one damage curve used in the analysis?

Answer: The vulnerability functions within the Aon IF model estimate the likely damage to a risk at a given wind speed. The input to the vulnerability function is the gust speed and the output is the loss as a percentage of the total insured value (TIV), known as the damage ratio (DR). The vulnerability function is split into two components, the chance of loss (COL) and the conditional mean damage ratio (CMDR) which is the expected damage given that a loss is occurring. For a windstorm, the COL is low across most of the affected areas; if a postcode is hit by a 25 m/s gust, most of the buildings will not experience any loss.

The model uses damage matrices, wherein the hazard intensity is divided into bins, which are integer values of gust speed, and the estimated DR is divided into bins with each having a probability of being affected. Thereby, variation in the DR is considered. There will be a Figure illustrating this concept in the model documentation in the Supplementary of the revised manuscript.

Section 3.1: The 98th percentile acts quasi as a representation of building codes/standards for the LI approach. Does the AON model also have building code regions (where a similar wind speed would cause a different loss) implemented and if yes, how does the 98th "building code region" pattern compare to these?

Answer: Yes, there are 180 different vulnerability regions in Europe, where unknown building typology damage curves are based on the available building index. These vulnerability regions reflect differences in the engineering vulnerability due to the local building inventory.

Section 3.2

1. Table1: Correlation numbers for the loss seem to be mis-aligned with Figure 3, e.g. France 0.9059 vs 0.62

Answer: Thanks for pointing this out. Based on the comments by reviewer 2, we decided to merge Tables 1 and 2, and to replace the numbers with the $R^2$ values from the Spearman's rank correlation (see Table RC3.1).

*Table RC3.1:* R² of Spearman's rank correlation coefficient between LI ERA5 and LI ERA-Interim (2nd column), LI ERA5 and Aon's IF Euro WS model (3rd column), LI ERA5 and PERILS (4th column), and Aon's IF Euro WS model and PERILS (last column). The number of common storms per country is given in brackets.

| | LI ERA5 vs LI ERA-Interim | LI ERA5 vs Aon's IF Euro WS | LI ERA5 vs PERILS | Aon's IF Euro WS vs PERILS |
|---|---|---|---|---|
| **Core Europe** | 0.65 [20] | 0.52 [23] | 0.26 [17] | 0.57 [19] |
| **Austria** | 0.43 [20] | 0.75 [15] | 1.0 [4] | 1.0 [4] |
| **Belgium** | 0.62 [20] | 0.22 [21] | 0.09 [11] | 0.66 [11] |
| **Denmark** | 0.25 [20] | 0.41 [15] | 0.49 [5] | 0.14 [6] |
| **France** | 0.79 [20] | 0.6 [17] | 0.56 [10] | 0.54 [11] |
| **Germany** | 0.5 [20] | 0.57 [23] | 0.33 [15] | 0.47 [15] |
| **Ireland** | 0.37 [20] | 0.2 [19] | 0.49 [5] | 0.64 [5] |
| **Luxembourg** | 0.64 [20] | 0.26 [15] | 0.07 [6] | 0.43 [6] |
| **Netherlands** | 0.2 [20] | 0.64 [21] | 0.68 [11] | 0.7 [11] |
| **Norway** | 0.29 [20] | 0.4 [9] | 0.25 [3] | 1.0 [3] |
| **Sweden** | 0.51 [20] | 0.23 [13] | 1.0 [4] | 0.16 [4] |
| **United Kingdom** | 0.49 [20] | 0.36 [20] | 0.44 [13] | 0.7 [13] |

2. Ln205: "unlike ERA-interim, ERA5 shows a broader area of high wind guest, especially over the UK and Western continental Europe". Despite, the outlier shows very low LI ERA5 values?
Answer: The footprint for Irina is overall flatter in ERA5 compared to ERA-Interim. This is particularly the case for the UK, where the mean wind gust over land is 12.1 m/s for ERA5 and 24.6 m/s for ERA-Interim. Therefore, the LI for ERA-Interim is higher due to the cumulative effect (summation of $v/v_{98}$). Additionally, Figure RC3.6 shows an extension of Figure S1 including another panel showing the ERA5 footprint re-gridded to the ERA-Interim resolution. This confirms the overall flatter structure of footprint. We agree this is an important point and we will add more details and Figure RC3.6 in the revised manuscript.

[Figure]

*Figure RC3.6:* Wind gust footprint for storm Irina in October 2002 based on ERA5 (a), ERA5 re-gridded to the ERA-Interim grid (b), and ERA-Interim (c). Shown is the largest exceedance (in percent) of the 98th percentile of

*daily maximum wind gust within 72 hours. The red line and dots denote the cyclone track derived from ERA5 (a, b) and ERA-Interim (c) using the tracking algorithm of Pinto et al. (2005).*

3. Ln215: The obvious reason is the higher spatial resolution of ERA 5: as stated above: to make the analysis more revealing, it is suggested to aim to remove the grid square # dependency by normalization/scaling in the LI formula (1) and look at the remaining differences. Then also absolute values can be used in the analysis of Figure 3.
   Answer: Please refer to our reply on the first comment in section 2.1

4. Ln219: ranking of storms: Does this rank analysis really add significant value? At least some rationale for the differences should be given. Consider to instead translate the LI index values in monetary amounts by using one recent storm as reference loss/by normalizing it with e.g. PERILS loss estimate. This would be more tangible and can be also used to compare to AONs estimates (if available) in section 4.
   Answer: We think the analysis of the ranked storms is meaningful, as it actually demonstrates the strengths and usefulness of the LI methodology. Previous studies like Leckebusch et al. (2007) tried to translate loss from a meteorological index into monetary values, based on a very simple linear regression. However, the LI approach is always much simpler, as – unlike the Aon IF model – it does not have an exposure, vulnerability or economic component. Since our study is the first to compare a full insurance windstorm model (which is not available publicly) to a simplified meteorological loss index (which only considers the hazard component), we focus thus on a straightforward comparison of the two methods. Therefore, in our opinion, the objective should not be to translate the LI values in monetary values, but to compare the two methods while considering the Aon IF model output as a representation of real loss. Finally, we aim to extract meaningful conclusions about the similarities and differences, strengths and weaknesses.

Section 4: Table2: Suggest to add the number of storms for the ERA5 data set to allow for a comparison to the AON model

Answer: We are not sure if we understand this comment. The number of storms in ERA5 depends on our definition. As we are interested in extreme storms, we consider only events above a certain threshold, which corresponds to an average of five events per season. Therefore, the event set of ERA5 contains 205 events (=41 seasons*5events). For the comparison, we only focus on common events.

Section 4.1.

1. The core analysis of the paper (as also stated in the title) is to reveal the differences between the LI and AON model loss estimates which is mainly addressed in a scatter plot analysis. It is obvious that there are substantial differences between the two approaches. More emphasis should be put on revealing the reasons for the differences seen. Most likely this is due to the different vulnerability curve shape of the v3 approach and the AON model. Since the v3 approach was used in literature quite often in the past, it would be very beneficial to work out the limitations and suggest improvements based on the learnings of the comparison.

However, this would likely mean to have more insights in the AON approach and to cross-check with real world loss numbers.

Answer: Please refer to our replies to the general comments above. Additionally, we would like to mention that the LI method used in our study is well established. Based on the original approach developed for station data from Klawa & Ulbrich (2003), it was further developed by Pinto et al. (2012), and several formulations (also with other exponents) were tested. The same was done in other studies such as Prahl et al. (2015). All these studies agree that the performance of the different indices depends on the underlying event set. For some storm events, formulations with higher exponents seem to better suit to realistically estimate windstorm losses, while for other events, the cubic relationship provides results that are more realistic. In this sense, and based in our experience, no formulation clearly outperforms the others. Since our study is the first to compare a full insurance windstorm model (which is not available publicly) to a simplified meteorological loss index, we focus on a straightforward comparison of the two methods. Nevertheless, we will provide a more detailed discussion of the impact of the LI setup on the results in the revised manuscript.

2. Figure 6 shows clearly the very different behavior of both approaches. Maybe a more in depth analysis for some few selected storms can help to shed more light on the differences, e.g. by looking into the loss contribution to the overall loss by wind speed.

Answer: Please refer to our reply on the previous comment.

Section 5

1. "For all of Europe, LI values are higher for ERA5 than for ERA-interim". This is mostly an effect of different horizontal resolution. Suggest to remove and discuss the residual differences as stated above.

Answer: Please refer to our reply on one of the previous comments.

2. "Compared to AON's IF model, LI ERA5 shows overall lower loss values". This statement is only true for normalized values but not for (more relevant) monetary values. For this (highly valuable) analysis, LI loss index values need to be translated in monetary loss values and compared to AON's output.

Answer: Please refer to our comment on section 3.2

3. Ln296: "the AON model seems to better distinguish between high and moderate impact events": Without benchmark with real world loss numbers this statement appears quite subjective.

Answer: We agree with the reviewer in this point. The Aon IF model is calibrated against the PERILS dataset, thereby it can be assumed as a representation of a "market-reality" for the purpose of this paper. Nevertheless, we decided to use none of the datasets as ground truth (Moemken et al., 2024). We will carefully go through section 5 and rephrase sentences where necessary.

4. Ln 299: …, the catastrophe model shows a clear regional dependency of loss values. This regional dependence in less pronounced in LI ERA5. Suggest to discuss the reason for this

behavior. Are AON footprints downscaled or have higher resolution?

Answer: The hazard resolution in the Aon model is 7 km. Please also refer to the replies on the other comments regarding the Aon model.

5. Ln316: As stated above, what is the impact of using a 24h definition also for the LI approach?

Answer: Please refer to our reply on one of the previous comments.

6. Ln333: "the LI index is missing a detailed damage component, thus struggles to capture the non-linear response of the buildings at the tail of the gust spectrum for high impact events". The LI v3 approach is obviously non-linear, so the question is what stronger non-linearity would be more suitable. But foremost, from the analysis it is only clear how the wind-loss relationship compares to the unknown AON approach and it is difficult to draw conclusion on the quality of the approaches without comparison to reality.

Answer: We agree with the reviewer and will reword the statement in the revised manuscript. The Aon IF model is calibrated against real loss data, using the PERILS data as the primary benchmark. Therefore, all storms considered in our analysis are calibrated/validated against a "market-reality". We will clarify this in the revised manuscript. Additionally, we will include the PERILS data in some of the analysis (see above).

7. Ln343: "…it is suitable for estimating the impact…" Without more quantitative measure and real world comparisons (and given the large differences to the supposedly more sophisticated AON approach) it is hard to follow this conclusion.

Answer: We will reword the statement in the revised manuscript (see also previous comments).

**References**

Feser, F., Barcikowska, M., Krueger, O., Schenk, F., Weisse, R., and Xia, L.: Storminess over the North Atlantic and northwestern Europe – A review, Quart. J. Roy. Meteor. Soc., 141, 350-382, https://doi.org/10.1002/qj.2364, 2015

Karremann, M. K., Pinto, J. G., von Bomhard, P., and Klawa, M.: On the clustering of winter storm loss events over Germany, Nat. Hazards Earth Syst. Sci., 14,2041-2052, https://doi.org/10.5194/nhess-14-2041-2014, 2014

Klawa, M. and Ulbrich, U.: A model for the estimation of storm losses and the identification of severe winter storms in Germany, Nat. Hazards Earth Syst. Sci., 3, 725-732, https://doi.org/10.5194/nhess-3-725-2003, 2003

Leckebusch, G. C., Ulbrich, U., Fröhlich, L., and Pinto, J. G.: Property loss potentials for European midlatitude storms in a changing climate, Geophys. Res. Lett., 34, L05703, https://doi.org/10.1029/2006GL027633, 2007

Moemken, J., Messori, G., and Pinto, J. G.: Windstorm losses in Europe – What to gain from damage datasets, Weather and Climate Extremes, 44, 100661, https://doi.org/10.1016/j.wace.2024.100661, 2024

Pinto, J. G., Karremann, M., Born, K., Della-Marta, P., and Klawa, M.: Loss potentials associated with European windstorms under future climate conditions, Climate Res., 54, 1-20, https://doi.org/10.3354/cr01111, 2012

Prahl, B. F., Rybski, D., Burghoff, O., and Kropp, J. P.: Comparison of storm damage functions and their performance, Nat. Hazards Earth Syst. Sci., 15, 769-788, https://doi.org/10.5194/nhess-15-769-2015, 2015

---

## Author Comment (AC4)

We thank Mathias Raschke for his comments on our manuscript.

I am very interested in any kind of quantification of aggregated natural catastrophes (natcat) losses from such as you present, as I am employed in (re)insurance industry for natcat modelling and research and publish as freelancer in this field. Unfortunately, I realize many gaps and shortfalls in and questions regarding your draft about event losses from windstorms (in parts of Europe):

- Why is the renaming (storm severity index [SSI] to meteorological loss index [LI]) not mentioned in the abstract?
  Answer: We did not rename the method, but used the official definition/naming by Pinto et al. (2012) – a well-established method. LI was introduced to differentiate between a formulation using population density as a proxy for the insurance values (LI) and another one without, thus representing the purely meteorological effect (MI).

- As far as I know, the original purpose of the SSI was not to estimate loss, but to formulate a size measure (for spatial extent (RMS 2024)). The SSI is also used to quantify spatial correlation (Bonazzi et al. 2012).What means a "metrics to quantify windstorm-related losses"? An event loss can be recorded or estimated. Why so you change this focus?
  Answer: The original papers to formulate the SSI approach were Palutikof & Skellern (1991) and Klawa & Ulbrich (2003). In our study, we follow the approach from Klawa & Ulbrich (2003) and Pinto et al. (2012) focusing on the aggregated loss. Even the above-mentioned study by Bonazzi et al. (2012) states that "The SSI is a hazard-based index which correlates closely to aggregated damages due to storms." They use it "for its relevance to the re/insurance industry." Therefore, in our opinion, we did not change the focus of SSI.

- The natcat models in (re)insurance industry (such as the applied Aon's IF Euro WS model) are less explained in the draft in contrast to your reference Mitchell-Wallace et al. (2017). Alternative models and vendors are not even mentioned. The Natcat model with thousands of stochastic events estimates event losses for a defined exposure and (high) return period.
  Answer: We will provide a more detailed description (roughly 10 pages) of the Aon IF windstorm model, which will be included in the Supplementary. Please also refer to the main reviewers' comments.

- What sense does it make to compare two model results on event losess when data for (insured) event losses and corresponding exposure (provided by the Perils AG, mentioned in your draft) are available? There is also information about market penetration (proportion of insured exposure to insurable exposure).
  Answer: Our study is the first to compare a full insurance windstorm model (which is not available publicly) to a simplified meteorological loss index. For proprietary reasons, our analyses are restricted to country-scale and normalized losses for the Aon model output. Nevertheless, we will include a detailed case study for recent storm Sabine (February 2020) in the updated manuscript, in which we compare LI and the Aon model output, and add aggregated market losses from the PERILS data as a reference.

- Why is my model (Raschke 2022) not even mentioned? The agreement between estimated and observed event damage (windstorm Germany) is significantly better in my results (plot) than in yours.
  Answer: Thanks for making us aware of this study; we will include a reference to it in revised manuscript.

- The loss/damage function applied should be discussed. The power parameter of 3 might be unrealistic. The wind speed does not cause damage, but rather it creates a damage generating wind pressure/load (at the buildings) that is proportional to the squared wind speed (details see Raschke 2022).

  Answer: The LI method used in our study is well established. Based on the original approach developed for station data from Klawa & Ulbrich (2003), it was further developed by Pinto et al. (2012), and several formulations (also with other exponents) were tested. The same was done in other studies such as Prahl et al. (2015). All these studies agree that the performance of the different indices depends on the underlying event set. For some storm events, formulations with higher exponents seem to better suit to realistically estimate windstorm losses, while for other events, the cubic relationship provides results that are more realistic. In this sense, and based in our experience, no formulation clearly outperforms the others. Since our study is the first to compare a full insurance windstorm model (which is not available publicly) to a simplified meteorological loss index, we focus on a straightforward comparison of the two methods. Therefore, in our opinion, the objective should not be to experiment with the LI formulation. Nevertheless, we will provide a more detailed discussion of the impact of the LI setup on the results in the revised manuscript.

- Correlation measures has been formulated for random variables. The maximum event loss per year or the annual sum of such losses are random variables (drawn once per year). However, event losses don't be random variables but point events of a stochastic process. Therefore, you can't just apply a correlation measure to it.

  Answer: We are sorry, but we cannot follow the reasoning here.

Besides, the results are not perfectly presented (e.g., Figure 8, colour scale for correlation measure form blue [-0.9] to red [0.9] although only positive correlations are mapped).

Answer: We will improve the presentation of results in the revised manuscript.

References

Klawa, M. and Ulbrich, U.: A model for the estimation of storm losses and the identification of severe winter storms in Germany, Nat. Hazards Earth Syst. Sci., 3, 725-732, https://doi.org/10.5194/nhess-3-725-2003, 2003

Palutikof, J. P. and Skellern, A. R.: Storm severity over Britain: a report to Commercial Union General Insurance, Climatic Research Unit, School of Environmental Science, University of East Anglia, Norwich, UK, 1991

Pinto, J. G., Karremann, M., Born, K., Della-Marta, P., and Klawa, M.: Loss potentials associated with European windstorms under future climate conditions, Climate Res., 54, 1-20, https://doi.org/10.3354/cr01111, 2012

Prahl, B. F., Rybski, D., Burghoff, O., and Kropp, J. P.: Comparison of storm damage functions and their performance, Nat. Hazards Earth Syst. Sci., 15, 769-788, https://doi.org/10.5194/nhess-15-769-2015, 2015

---

## Author Comment (AC5)

We thank Irene Garcia Marti for her detailed comments.

Dear authors, I read your paper with curiosity. I think it is interesting for scientists to begin collaborating with the insurance sector, so that researchers and insurers can better understand the impacts of severe weather on different socio-economic sectors. I do think it is a timely topic, but I believe the manuscript requires quite some work to become clear and deliver its core messages effectively. Hence, my recommendation is a Major Revision, and I hope the comments in this document will be helpful. Good luck.

**Major comments**

In this section you can find comments in two categories: structural and data analysis. For the structure, I have the impression the paper could benefit from a clearer structure, with a better division between data descriptions and the methods, whereas for the analysis comments there are parts that remain unclear.

1) Structural comments

Introduction: The introduction requires some streamlining, since it intertwines motivating reasons to carry out such a study with lengthy descriptions of previous work. As a result, it is difficult to follow the storyline the authors wish to convey. For example, in L36-L51 you begin talking about the hazard, exposure, vulnerability framework, but this somehow becomes diluted in the rest of the paragraph. It might be helpful for readers to center the introduction about these three components of risk management using the risk propeller figure, so that the references to these multiple insurance companies and other articles are somehow anchored to this image. Then in L65-L68 the authors roughly describe the analysis that will be doing, which I find too detailed for an introduction, to then explain the paper structure, which jumps back to the general scope. Overall, I think this section requires streamlining and making sure the message the authors wish to convey is effectively delivered.

Answer: We thank the reviewer for her suggestion. We will streamline the introduction by shortening the paragraph on damage datasets and expanding the part on damage functions. This is also in line with the comments by the other reviewers.

Data and methods: I would recommend re-structuring this section. While reading, there are parts mixing data description with the methods, which interrupt the flow. For example, L102-L114 describe the ERA5 data (and other generalities) right after the equations for LI are presented. Then in L116 the flow is recovered. Same goes for the description of PERILS in L154-L163. On the one hand, in the introduction the authors mention a hazard-exposure-vulnerability schema. On the other hand, I have the impression that the hazard component is ERA5, the exposure is PERILS, and the vulnerability the data/curves from AON. So I would recommend restructuring this section in 2.1.1 - Hazard; 2.1.2 - Exposure; 2.1.3 - Vulnerability and then a 2.2 - Meteorological loss index and 2.3 - Catastrophe model that are thoroughly explained without data description intrusions.

Answer: We will improve the structure of section 2 by clearly separating the data description from the methods.

Summary and discussion: I find this section long and I am not sure what the main conclusions of this work are. Is there any way of separating the "more technical" discussion part from the "more abstract" conclusions? Overall, I do not see the "take home message", or how does this relate with the two very concrete research questions posed in L60-L64. Also, how might the insurance sector be using the insights gained in this study?

Answer: We will improve the structure of the discussion, also considering the other reviewers' comments.

2) Data analysis comments

*2.1 - Meteorological loss index*

In L86 the text say "Losses are proportional to the wind power or the wind kinetic energy flux....". Perhaps softening or extending this description might be useful for a generic reader to comprehend the meaning and implications of this.

Answer: This is the standard explanation for justifying the cubic relationship used in the LI calculation (see e.g. Klawa & Ulbrich, 2003). We will try to formulate the sentence more clearly in the revised manuscript.

In L87-L88 you mention that "...only the 2% of wind gusts....cause damage". I am missing here some elaboration about what are the damages that you have in mind. Are we talking infrastructural damage? Agricultural damage? To public or private assets? Is personal propriety included here? If this is one of the four assumptions in the paper, I would expect to have a solid description of what is the meaning of "damage" for the authors in this work.

Answer: We mean private buildings, in line with the reasoning by Klawa & Ulbrich (2003). We will clarify this in the updated manuscript.

In L90 you mention "In the case that no insurance data, population density can be used as a proxy for the exposure component". Indeed, but then does it mean that you are focused in damage in cities, hence, roads, agriculture, or forestry damages are out of the study? Also, how frequently do you bump into records that have no insurance data associated? I think this study could benefit from some extra clarity on how much insurance data is available, as long as its contents. This might help at assessing whether population density is a matching candidate for the insurance data or requires combining it with other layers (e.g. land use, urban tree, urban morphology).

Answer: As mentioned in the previous reply, we focus only on private buildings at a scale above city level, and thus in an aggregated form (here: 0.25°x0.25°). We will clarify this in the revised manuscript.

Also, I wonder how the different spatial dimensions are accommodated in this analysis. For example, population density from CIESIN at 0.25deg is roughly 30km, but then how insurance data are aggregated? Per country? Per NUTS region? And how does this relate with the spatial resolution of ERA5, ERA5-Interim and the catastrophe model from Aon? I believe it would be useful to have a section discussing the harmonization of the spatial dimension, so that it is clearer what the two models receive as input.

Answer: Our study is the first to compare a full insurance windstorm model (which is not available publicly) to a simplified meteorological loss index. For proprietary reasons, we can only use the Aon model output at country-scale and in a normalized (and thereby anonymized) form. Therefore, we also aggregate the LI at country level and focus on a straightforward comparison of the two methods. We will explain this better in the revised manuscript.

*4.1 - Windstorm loss*

Here in Figure 5 some results are visualized in the geographic space. In this figure I have two comments. First, the results are presented in a per-country basis, but the analysis seems to have been carried out on pixels much smaller than the country surface. I wonder if results can be

presented using NUTS 2 regions or a spatial unit that is closer to the spatial dimension of the analysis. If results are aggregated for the sake of visualization, this would be understandable, but then I would expect a clearer description of the treatment of the spatial dimension throughout the manuscript. What is the resolution of the insurance data? How are all these harmonized? Second, the colorscale chosen in this figure might not be ideal to visually perceive differences. Perhaps a sequential colormap (with 3 colors) or a perceptually uniform sequential colormap (eg. Like viridis) might be a better choice to guide the reader to the differences you describe.

Answer: The reviewer is correct that the LI calculation is based on gridded data. However, as stated in the previous comment, the Aon model output is only available at country level. Therefore, we decided to aggregate all data to the same "spatial units" (= countries). In the revised manuscript, we will change the coloring of some figures to better highlight differences.

Also, I do not really understand how to interpret the Figures with the storm ranks. What helps the reader understand what is relevant?

Answer: The idea behind the Figures of the storm ranks was to compare the storm ranking between ERA5 (LI method) and Aon IF's Euro WS model. The results are straightforward to understand: If the $R^2$ is close to one, then there is a good agreement between the rankings computed with different methodologies. Lower $R^2$ means a higher disagreement between the two datasets. In the revised manuscript, we will additionally show Spearman's rank correlation coefficients at country level to improve the presentation of these results.

**Minor comments**

Answer: Thanks for the comments; we will consider them in the revised manuscript.

References

Klawa, M. and Ulbrich, U.: A model for the estimation of storm losses and the identification of severe winter storms in Germany, Nat. Hazards Earth Syst. Sci., 3, 725-732, https://doi.org/10.5194/nhess-3-725-2003, 2003